# MS-Bench: Evaluating LMMs in Ancient Manuscript Study through a Dunhuang Case Study

**Yuqing Zhang**[1]* **Yue Han**[1]* **Shuanghe Zhu**[1] **Haoxiang Wu**[1] **Hangqi Li**[1]
**Shengyu Zhang**[1]† **Junchi Yan**[2] **Zemin Liu**[1] **Kun Kuang**[1] **Huaiyong Dou**[1]†
**Yongquan Zhang**[1] **Fei Wu**[1]†
[1]Zhejiang University    [2]Shanghai Jiao Tong University
Project page: https://github.com/ianeong/MS-Bench

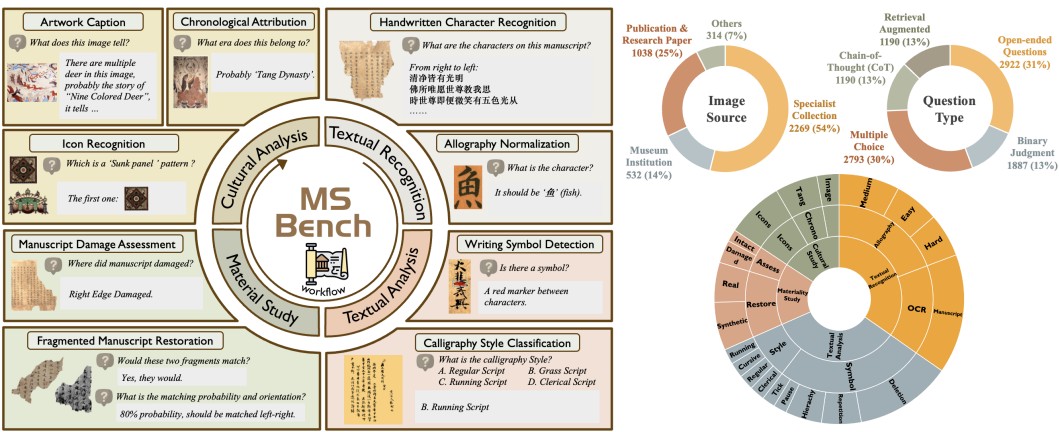

Figure 1: Overview of MS-Bench. Our comprehensive benchmark consist of multi-source, multi-scale manuscripts. Tasks are hierarchically organized to address domain challenges. Multiple question formats assess LMMs' effectiveness and robustness.

## Abstract

Analyzing ancient manuscripts has traditionally been a labor-intensive and time-consuming task for philologists. While recent advancements in LMMs have demonstrated their potential across diverse domains, their effectiveness in manuscript study remains underexplored. In this paper, we introduce MS-Bench, the first comprehensive benchmark co-developed with archaeologists, comprising 5,076 high-resolution images from 4th to 14th century and 9,982 expert-curated questions across nine sub-tasks aligned with archaeological workflows. Through four prompting strategies, we systematically evaluate 32 LMMs on their effectiveness, robustness, and cultural contextualization. Our analysis reveals scale-driven performance and reliability improvements, prompting strategies' impact on performance (CoT has two-sides effect, while visual retrieval-augmented prompts provide consistent boost), and task-specific preferences depending on LMM's visual capabilities. Although current LMMs are not yet capable of replacing domain expertise, they demonstrate promising potential to accelerate manuscript research through future human–AI collaboration.

---

*Equal contribution. † Corresponding authors.

39th Conference on Neural Information Processing Systems (NeurIPS 2025) Track on Datasets and Benchmarks.

# 1 Introduction

Ancient manuscripts serve as invaluable witnesses to human civilization, preserving a wealth of first-hand historical records of cultural traditions, economic activities, scientific advancements, and artistic evolution. The study of these documents is inherently interdisciplinary, integrating *palaeography* which focuses on character and textual analysis, *codicology* which examines physical restoration, and *iconography* which explores visual elements such as illustrations and murals [19]. Traditionally, these fields have relied on manual approaches facing several challenges: "inefficiency" due to repetitive and time-consuming tasks, "collaboration gaps" as experts specialize in their own domain, "subjective bias" that research results may depend on the expertise and experience of individual scholar [35].

To mitigate these issues, specialized AI models have been developed for manuscript recognition, restoration, and decipherment across various civilizations, from ancient Greek inscriptions and Mediterranean scrolls to Asian manuscripts [7, 22, 31, 50, 52]. Although these applications demonstrate notable progress in automating repetitive workflows, three fundamental limitations persist in current **task-specific** approaches. (1) **Data Scarcity:** existing domain-specific models rely on extensive expert-annotated training data, yet synthetic datasets may fail to capture paleographic diversity accurately. (2) **Cross-Disciplinary Integration Barrier:** complex analytical tasks (e.g., fragment restoration and chronological attribution) require synergistic reasoning across domains, while single-task models struggle to unify visual-language and cross-disciplinary knowledge. (3) **Generalization Constraints:** models optimized for specialized tasks are tightly coupled with domain priors, limiting multi-task generalization or adaptation beyond their pre-defined applications.

The emergence of Large Multimodal Models (LMMs), empowered by large-scale pre-training, enables end-to-end visual-language processing and zero-shot generalization, positioning them as theoretically viable solutions to the aforementioned challenges. However, their practical effectiveness in real-world archaeological research remains untested. First, *can current LMMs consistently assist scholars across diverse manuscript sources, adapting to real-world variability (**Operational Effectiveness**)?* Second, *what inherent limitations arise as task complexity increases and deeper cultural contextualization is required (**Capability Boundary**)?* Third, *how does performance fluctuate across different instruction types, from direct perception queries to multi-step reasoning and visual retrieval-augmented prompts requiring cross-modality knowledge integration (**Instructional Robustness**)?* Thus, a systematic evaluation of LMM effectiveness and reliability in historical document studies is essential, providing insights into their strengths, weaknesses, and areas for improvement.

To address these challenges, we introduce **MS-Bench, the first comprehensive benchmark for ancient manuscript analysis.** Curated from over 15,000 digitized manuscripts spanning the 4th to 14th centuries, MS-Bench comprises 5,076 high-resolution images alongside 9,982 task-specific questions, capturing diverse document types across 7 historical periods. As illustrated in Figure 1, our data sources extensively cover scholarly research publications over past 40 years (e.g., monographs, journal articles, etc.) and high-quality digitized archives from the past decade (e.g., British Museum, National Museum of China, etc.). Through a three-year interdisciplinary collaboration involving 12 senior researchers across 18 panels, we structured MS-Bench along *Research Workflows* reflecting real-world archaeological methodologies and *Cognitive Hierarchy* progressing from shallow character-level perception to holistic cultural reasoning. To ensure a rigorous evaluation of LMM capabilities, we designed four specialized prompting strategies, including direct Q&A, Multiple-choice, Chain-of-Thought (CoT) Reasoning, and Retrieval-Augmented Visual Context, to systematically assess LMM's effectiveness, robustness, cultural knowledge grounding and cross-image reasoning. Additionally, human experts validated benchmark data quality and provided manual baselines, enabling a direct comparison between human and model-generated responses. By integrating computational evaluation with manuscriptology's disciplinary depth, MS-Bench establishes a new standard for assessing LMMs on complex, historically grounded, and previously unseen manuscript tasks.

Our systematic evaluation of 14 closed-source, 18 open-source and 2 reasoning LMMs reveals key insights into their capabilities and limitations:

**(1) LMMs as assistive tools for archaeologists.** LMMs excel in standardized tasks such as allograph recognition and calligraphy style classification, effectively reducing labor-intensive workflows. However, they remain insufficient for expert-level manuscript discovery, necessitating domain-specific adaptation and enhanced multimodal alignment for cultural knowledge-intensive tasks.

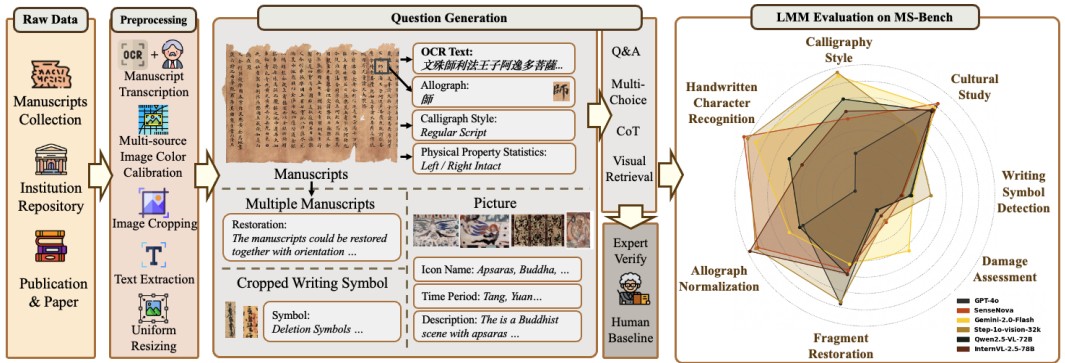

Figure 2: Illustration of MS-Bench construction pipeline (data source collection, preprocessing, question generation, annotation and human expert verification) and LMM evaluation results. LMMs demonstrate task-specific capability divergence.

**(2) LMMs demonstrate scale-driven reliability.** Small LMMs (<10B parameters) hallucinate more frequently on unfamiliar tasks, producing plausible-but-incorrect guesses due to overconfidence. In contrast, larger LMMs exhibit calibrated self-awareness, explicitly admitting knowledge gaps. Additionally, closed-source LMMs (e.g., GPT-4o, Gemini, etc.) outperform open-source models in instruction adherence (structured outputs & stepwise CoT) and visual comprehension, benefiting significantly from retrieval-augmented prompts for analogical reasoning and knowledge extrapolation.

**(3) CoT has a two-sided impact on different multimodal tasks.** On the positive side, CoT mitigates modality bias by realigning LLMs toward visual inputs and reducing their over-reliance on textual priors. It also helps distinguish genuinely knowledge-based reasoning from random guessing. However, on the negative side, CoT improves performance in fine-grained visual task (symbol detection) but hinders tasks requiring holistic perception (calligraphy style classification and chronological attribution). Even SOTA multimodal reasoning models (Claude-3.7, QVQ) may fail to demonstrate clear advantages in MS-Bench tasks. Unlike CoT, retrieval-augmented strategy provides a more stable performance improvement, with variation depending on model's capabilities.

**(4) Task-Driven Capability Divergence.** Our benchmark reveals that LMMs exhibit distinct specialization patterns: GPT-4o excels in instruction fidelity and generalist problem-solving, but struggles with ancient Chinese character-level recognition. Gemini demonstrates strong cross-image relational reasoning yet lacks pre-trained cultural contextualization. Step-1o-Vision-32k shows the most extensive cultural knowledge reserves, but fully unlocking its potential requires careful task-specific prompting. These findings highlight that no single model universally excels across all tasks. Instead, LMM selection should be guided by specific requirements, balancing general adaptability, visual-semantic precision, and cultural knowledge depth.

## 2 MS-Bench

### 2.1 Design Principle

Manuscript studies present distinctive computational challenges: their *scholarly requirements* demand both technical precision and historical sensitivity, their *task granularity* ranges from fine-grained character-level analysis to holistic interpretation of entire scrolls, and their *data diversity* spans from eroded textual fragments to pictorial artifacts. MS-Bench is designed to bridge the gap between comprehensive LMM evaluation and the intrinsic challenges of manuscript research. We adhere to the following three principles:

**(1) Scholarly-driven Holistic Task Design Philosophy:** MS-Bench encapsulates archaeologists' workflows, from labor-intensive, time-consuming and error-prone process in *Textual Recognition & Analysis*, to context-intensive reasoning in *Materiality & Cultural Study*.

**(2) Hierarchical Task Framework:** Co-developed with 7 domain experts, MS-Bench categorizes tasks into *4 vertical tiers* and *9 horizontal sub-tasks*, as shown in Table 1, progressively increasing in

Table 1: Details of our **MS-Bench**. Subsequent processing and annotation are detailed in Appendix B.

| Category | Task | Data Format | Quantity | Avg. Size |
|---|---|---|---|---|
| Textual Recognition | Handwritten Character Recognition | Original Manuscript | 778 | 1098×1254 |
| | Allograph Normalization | Cropped Single Characters | 970 | 322×328 |
| Textual Analysis | Writing Symbol Detection | Cropped Manuscript | 521 | 233×600 |
| | Calligraphy Style Classification | Cropped Manuscript | 400 | 470×571 |
| Materiality Study | Manuscript Damage Assessment | Original Manuscript | 332 | 937×823 |
| | Fragmented Manuscript Restoration | Multi-Original Manuscripts | 854 | 629×607 |
| | Icon Recognition | Multiple Images | 900 | 619×549 |
| Cultural Study | Chronological Attribution | Single Image | 276 | 446×630 |
| | Artwork Caption | Single Image | 45 | 764×708 |

contextual dependency and interpretative depth. To further evaluate LMM adaptability, we incorporate diverse question formats, simulating the layered inquiry process employed by human scholars.

**(3) Large-scale Multi-source Data Curation:** Centered on the most extensive and diverse collection of Dunhuang manuscripts, MS-Bench integrates 5,076 high-resolution images and 9,982 Q&A pairs. It incorporates decade-long accumulated research, public materials, and academic publications capturing stylistic evolution. The high-quality dataset provides a comprehensive and reliable foundation for benchmarking LMMs in manuscript studies.

## 2.2 Benchmark Construction

**Data Source.** Manuscript research has historically been conducted through isolated case studies due to a lack of high-fidelity image collections. Only in recent years have early-stage digitization efforts been initiated across global museums. Given the inherent scarcity of large-scale open datasets with cross-modal annotations, we integrate a synergistic data source through task-oriented approach. The foundation is built upon our team's 40-year **Dunhuang specialist-annotated archival collections**, preserving fine-grained paleographic features that are often lost in bulk digitization. These include *allograph characters, cropped writing symbols and verified fragmented matches.* We aggregate **open-source institutional repositories** from International Dunhuang Programme [32], Dunhuang Documents Database [17] and 9 other global museum collections. From these sources, we select 2598 high-quality digitized manuscripts for *handwritten character recognition, calligraphy style classification and materiality studies.* To address culturally intensive tasks, we supplement with **academic publications and papers**. These scholarly-grounded corpora provide the semantic depth necessary for manuscript interpretation and visual-textual reasoning.

**Preprocessing.** Despite rigorous source selection, inherent data heterogeneity persists due to variations in museum preservation conditions and digitization methods. To ensure data consistency, we implement a unified preprocessing pipeline tailored to each task, including essential image scale normalization, color calibration, filtering of small and low-resolution images, super-resolution and contrast enhancement. The resulting dataset balances scholarly fidelity with computational robustness.

**Question Generation.** To replicate real-world human-model collaboration and assess LMM prompt robustness, we design task-specific question formats aligned with archaeologists' progressive inquiry process. Our question generation framework follows a **cognitive hierarchy principle**, progressing from intuitive perception to complex reasoning: **Open-ended Questions** allow intuitive free-form responses, serving as zero-shot baseline for visual perception and multimodal alignment. **Binary Judgment & Multiple-choice Questions** provide clear, structured answer choices, enabling exact-match scoring for factual accuracy evaluation. **CoT Prompts** require multi-step inference through structured reasoning paths. **Retrieval-augmented Visual Queries** assess unseen task adaptation using in-context examples, measuring LMMs' ability to perform visual analogical reasoning.

**Ground Truth Annotation.** We establish task-specific annotation protocols under domain specialist[2] supervision, for a balance between computational efficiency and scholarly rigor. For large-scale

---

[2]The specialists involved in ground truth data annotation, verification and human baseline experiments, hold Ph.D degrees with at least 5 years expertize in ancient manuscript philology. Each expert dedicate in a specific aspect, including text recognition, fragment reassembly, and cultural interpretation.

character annotation, we employ Rushi OCR[3] for initial transcription, followed by manual refinement through expert double-checking. The ground truth for tasks derived from our archived collection is fully annotated by experts. For the remaining tasks, ground truth annotations are extracted from authorized repositories and publications, initially processed by computer science students, and subsequently proofread by domain experts.

## 3 Experiment Results

We conduct extensive experiments on MS-Bench, evaluating 14 closed-source [2, 8, 21, 24, 39] and 18 open-source LMMs [1, 9, 14, 27–29, 34, 42, 47–49], and 2 advanced reasoning LMMs [5, 9]. Our evaluation follows MS-Bench's four-tiered hierarchical framework, analyzing model performance through task-specific metrics tailored to each category. We conduct controlled comparisons to assess the impact of different prompting strategies. To identify task-specific challenges and current LMM limitations, we perform in-depth failure case studies and error analysis, pinpointing areas where models struggle. Finally, we derive practical insights for model selection, outlining capability thresholds and bottlenecks in scholarly workflows. Following the practices in peer studies [10, 38], we utilize either official APIs for LMMs or standard deployments. To ensure deterministic evaluation, we fix hyperparameters and eliminate randomness. More details for experiment settings are available in Appendix C and D. We discuss here the common conclusions and insights obtained, and give more examples and cases in Appendix E- K.

### 3.1 Textual Recognition

Textual Recognition is a core competency in paleography and serves as a prerequisite for in-depth textual analysis, requiring LMMs to achieve both precise visual localization and robust optical character recognition (OCR) capabilities. For *Handwritten Character Recognition*, input manuscripts range from intact scrolls to fragments in both color and gray-scale conditions. We evaluate with **Accuracy** and **Levenshtein Edit Distance**, comparing LMM output against ground-truth transcriptions.

For the more nuanced *Allograph Normalization*, we present isolated character instances (cropped from larger manuscripts) for LMM to map historical allographs to their standardized modern equivalents via fine-grained visual perception. We categorize allographs to three tiers based on their frequency in ancient documents: easy, medium and hard. Performance is measured using **Accuracy**, evaluating the one-to-one correspondence between model predictions and ground truth labels.

**Discussion.** *Handwritten Character Recognition* reveals fundamental limitations in LMMs' OCR robustness when confronted with paleographic challenges: handwriting style variations, unconventional vertical layouts, and semantic shifts from modern language. As expected, all models underperformed relative to their general OCR capabilities. Closed-source LMMs exhibit greater adaptability, achieving 67.49% average accuracy (v.s. 51.57% for open-source models) and reducing edit distance by 2.5×. This performance gap stems from closed-source LMMs' superior visual localization and more accurate text bounding-box grounding. Additionally, they are less prone to repetition errors, a common issue observed in smaller models (e.g., InternVL2.5 series).

*Allograph Normalization* evaluates LMM's abilities to recognize and infer allograph through character structural similarities. Results reveal capability disparities: Step-1o-vision-32k and SenseNova lead with an impressive 73.34% average accuracy across all subsets. Interestingly, Qwen2-VL-2B outperforms larger models within its series, a phenomenon also observed in Qwen2.5-VL-3B's *Writing Symbol Detection* task. This deviation from typical scaling trends suggests that model size alone is not the primary determinant of performance in fine-grained recognition tasks. A detailed discussion on this anomaly is included in Appendix H.3.

Two key insights emerge from our analysis. First, LMMs could outperform untrained individuals in Allograph Normalization, even comparable to intermediate experts. Interestingly, when categorizing allograph based on philological principles, we find that LMMs excel in recognizing script variations (i.e., stroke additions or omissions) but struggle with semantic allographs involving radical substitutions. This suggests that current LMMs only *memorize* the glyphs instead of learning the intrinsic structural features or semantic information of ideograph characters, thus lack generalization.

---

[3]Specialized in ancient inscribed and handwritten texts, supporting multi-directional layouts of traditional Chinese characters: `https://guji.rushi-ai.net`

Table 2: Performance of LMMs on *Textual Recognition*. Only top-performance models are listed due to space limitation, complete results are available in Appendix. The best LMM of each set is **bold**, the second-best is underlined. "*N/A*" indicates such LMM does not accept multiple visual inputs. "*Fail*" indicates such LMM could not solve the task (refuse to answer, all return same answers or generate random guesses). These special notations are maintained in the following tables. For Human results, we randomly sample a 10% subset from MS-Bench.

| | Handwritten Character Recognition | | Allograph Normalization (Accuracy) | | |
| | Accuracy ↑ | Edit Distance ↓ | Easy ↑ | Medium ↑ | Hard ↑ |
|---|---|---|---|---|---|
| GPT-4o | *Fail* | *Fail* | 65.41% | 39.09% | 22.54% |
| Step-1o-vision-32k | 80.19% | 125.39 | **89.04%** | **77.16%** | **58.10%** |
| SenseNova | **82.88%** | 113.30 | 88.70% | 74.62% | 56.69% |
| Gemini-2.0-Pro-Exp | 76.53% | 78.54 | 61.64% | 43.91% | 33.80% |
| Gemini-2.0-Flash | 74.80% | **60.05** | 71.23% | 47.21% | 36.97% |
| Qwen2.5-VL-3B | 64.69% | 456.16 | 52.05% | 36.04% | 27.11% |
| Qwen2.5-VL-7B | **72.99%** | **92.63** | 61.30% | 42.39% | 29.23% |
| Qwen2.5-VL-72B | 69.39% | 102.99 | 55.82% | 39.85% | 26.76% |
| Qwen2-VL-2B | 28.40% | 325.51 | 77.74% | 56.85% | 38.73% |
| Qwen2-VL-7B | 46.40% | 425.25 | 44.86% | 29.70% | 25.00% |
| Qwen2-VL-72B | 51.78% | 236.80 | 30.14% | 16.75% | 11.27% |
| InternVL2.5-78B | *N/A* | *N/A* | **90.41%** | **81.98%** | **63.73%** |
| Valley-Eagle-7B | 34.96% | 581.91 | 80.48% | 58.83% | 40.85% |
| Untrained Human/Expert | *Fail* / 91% | *Fail* / 19.65 | 84% / 96% | 72% / 96% | 48% / 84% |

Second, we observed overall advancements with LMM version updates (Qwen2.5-VL surpasses Qwen2-VL by 26.83% in *Character Recognition*), in accordance to its generalized capability enhancement. A similar trend appears across other tasks, reinforcing that continuous LMM iterations (including scaling, improved training data and refined training methods) would significantly enhance both general and domain-specific performance. Specifically, we notice that the latest Qwen series models with dynamic resolution visual encoder handle irregularly shaped image inputs better. However, as we continuously increase image sizes and aspect ratios using manuscripts from British collection, LMM OCR performance significantly degraded. This suggests that dynamic resolution and more flexible image patching strategies may be a future direction for improving LMM performance.

## 3.2 Textual Analysis

Textual Analysis refers to the process by which paleographers transcribe, categorize, and analyze manuscripts, requiring both fine-grained vision-language alignment and broader feature extraction. These tasks are labor-intensive and context-dependent, posing significant challenges for LMMs.

***Writing Symbol Detection*** evaluates LMMs' ability to identify editorial and ritual marks, being critical for accurate transcription. Given a cropped manuscript region containing symbols or Buddhist mudra indicators, LMMs must verify symbol presence based on textual prompts (e.g., "Does fragment contain a circle annotation?"). Performance is measured by **Accuracy** across 4 question types.

***Calligraphy Style Classification*** categorizes manuscripts based on four dominant styles: Clerical Script, Regular Script, Running Script and Cursive Script. Models receive full-page manuscripts and classify styles based on stroke patterns. Performance is evaluated using **Accuracy**.

**Discussion.** *Writing Symbol Detection* presents LMMs with unique challenges beyond pre-trained common object detection and feature extraction: recognizing small, ambiguous symbols amid ink degradation and texture noise. Among all models, Gemini-1.5-Pro achieves the highest average accuracy of 43.27%, maintaining consistent performance across all four questioning styles. Binary Judgment questions, when supplemented with explicit symbolic descriptions (e.g., "A Hierarchical Symbol is a red solid or hollow dot"), significantly mitigated hallucinations. CoT prompting improved accuracy by 2.99%, as it forced models to explicitly align step-wise textual shape descriptions with visual symbols, reducing both modality misalignment and random guessing (models were even allowed to answer "not found" rather than making irresponsible guesses). Visual Retrieval Augmentation provided the most substantial performance gains (an average of 14.57%), as example symbol images provided clearer visual references for symbol recognition. This effect was most pronounced in models with stronger visual capabilities. Gemini-2.0-Pro-Exp achieved a 1.65×

Table 3: Performance of LMMs on *Textual Analysis*. Best in **bold** and second-best in underlined.

| | Writing Symbol Detection (Accuracy) | | | | Calligraphy Style Classification (Accuracy) | | |
| | Q&A ↑ | Binary ↑ | CoT ↑ | Retrieval ↑ | Q&A ↑ | CoT ↑ | Retrieval ↑ |
|---|---|---|---|---|---|---|---|
| Step-1o-vision-32k | 24.90% | **69.29%** | 26.85% | **40.94%** | 67.33% | 77.67% | 87.00% |
| Gemini-2.0-Pro-Exp | 22.73% | 67.95% | **40.12%** | 37.55% | 43.00% | 40.50% | 63.25% |
| Gemini-2.0-Flash | 6.72% | 55.09% | 20.55% | 34.98% | **75.33%** | 66.00% | 83.67% |
| Gemini-1.5-Pro | **29.05%** | 68.91% | 38.93% | 36.17% | 70.67% | 69.33% | 44.33% |
| Claude-3.5-Sonnet | 8.45% | 48.94% | 10.28% | 34.98% | 70.00% | 80.33% | 81.00% |
| GLM-4V-Plus | 0.97% | 51.06% | 0.39% | *N/A* | 65.00% | **80.67%** | 52.67% |
| Hunyuan-Vision | 3.95% | 33.78% | 9.49% | 10.33% | 74.33% | 63.00% | 56.67% |
| Hunyuan-Turbo-Vision | 1.38% | 31.67% | 8.89% | 5.34% | 60.67% | 76.00% | **90.00%** |
| Qwen2.5-VL-3B | **31.82%** | 39.35% | **22.13%** | 13.24% | 32.25% | 31.50% | 25.00% |
| Qwen2-VL-72B | 12.65% | 63.53% | 13.24% | 25.89% | 60.00% | 49.00% | 53.33% |
| InternVL2.5-8B | 2.72% | 46.45% | 19.26% | 20.23% | 68.33% | 45.33% | *Fail* |
| InternVL2.5-78B | 4.09% | 52.21% | *Fail* | 32.49% | 63.33% | 51.33% | 44.00% |
| LLaVA-OneVision-7B | *Fail* | **70.44%** | 14.40% | 18.48% | 76.00% | 65.67% | 57.33% |
| Phi-3.5-vision | *Fail* | 68.33% | *Fail* | **39.30%** | 64.00% | 74.00% | 52.00% |
| Claude-3.7 | 3.84% | 50.67% | 7.49% | 29.17% | 58.50% | 61.00% | 66.00% |
| Untrained Human/Expert | 36% / 88% | 52% / 98% | *N/A* | *N/A* | 40% / 78% | *N/A* | *N/A* |

accuracy gains, indicating that visual augmentation compensates for pre-trained knowledge gaps. Notably, Phi-3.5-Vision, an open-source model, even surpasses closed-source LMMs with 39.30% accuracy through this optimized visual prompting. This analogy-driven reasoning format boosted performance by 2.31× for all LMMs, remains the most effective strategy for this task. As Writing Symbol Detection is a labor-intensive yet crucial step in transcription and deciphering, LMMs can serve as viable reference tools, achieving a 3× efficiency gain over human experts in execution time.

*Calligraphy Style Classification* reveals significant data bias in LMMs, particularly in temporal misalignment: older fonts, such as Clerical Script, achieved only 1.25% accuracy, whereas scripts closely resemble modern typography (e.g., Regular Script, 63.42% accuracy) were easier to classify. This disparity arises from LMMs' predominant exposure to modern typography during pretraining, a trend similarly observed in *Chronological Attribution*.

Notably, CoT had minimal impact on *Calligraphy* and *Chronological Attribution* ($<0.5\%$ average improvement). Moreover, half of the evaluated LMMs even exhibited a decline in accuracy, as deeper reasoning often led models to respond with "unknown" or default to a single prediction across all queries (e.g., predict "Regular Script" for all cases). By comparing these two tasks with *Writing Symbol Detection*, we identify a key pattern: **Excessive CoT reasoning is detrimental to tasks requiring intuitive, holistic stylistic perception.** While CoT enhances fine-grained visual tasks, its effectiveness remains inferior to retrieval-augmented visual references. This is because CoT does not introduce new knowledge but rather reinforces pre-existing data biases and reasoning tendencies.

## 3.3 Materiality Study

Materiality Studies focus on analyzing the physical condition of ancient manuscripts, including damage assessment and multi-scale fragment restoration. *Damage Assessment* is a qualitative classification task where LMMs evaluate fragment degradation severity through multiple-choice queries. Performance is evaluated through **Accuracy**.

*Fragment Restoration* challenges models to infer contour and texture continuity in torn or eroded manuscripts. LMMs receive fragment pairs and predict whether they can be restored together, using: Binary Matching (direct yes/no judgment), Probabilistic Scoring (likelihood estimation on a 0-100 scale), orientation inference (relative positioning hypotheses with left/right/top/bottom). These tasks replicate the painstaking workflows, where manually matching puzzle-like fragments requires weeks of comparative analysis, relying on fiber texture, calligraphy style, content and contour similarity.

**Discussion.** *Damage Assessment* is a crucial pre-filtering step to reduce pairwise matching candidates in manuscript restoration. Models such as Gemini, GLM-4V-9B, and Qwen2.5-VL-7B exhibit moderate alignment ($\approx 43\%$ accuracy, compared to 25% random guess) when associating damage patterns with textual descriptions. However, most LMMs struggle with holistic integrity assessment, particularly when degradation occurs across multiple regions.

Table 4: Performance of LMMs on *Materiality Study*.

| | Damage Assessment(Accuracy) | Fragment Restoration (Accuracy) | | |
| | Multiple Choice ↑ | Binary ↑ | Probabilistic ↑ | Orientation ↑ |
| --- | --- | --- | --- | --- |
| GPT-4V | *Fail* | 64.20% | 58.70% | 49.20% |
| GPT-4o | 11.82% | 59.40% | 58.00% | **72.00%** |
| Step-1o-vision-32k | 19.31% | 62.50% | **60.00%** | *Fail* |
| SenseNova | 20.81% | **64.88%** | **66.23%** | 7.29% |
| Gemini-2.0-Pro-Exp | 40.89% | *Fail* | 49.60% | 42.20% |
| Gemini-2.0-Flash | **43.35%** | 49.8% | 48.80% | 48.80% |
| Qwen2.5-VL-7B | **43.84%** | *Fail* | 50.04% | 42.60% |
| Qwen2.5-VL-72B | 12.81% | 50.30% | 50.00% | **48.00%** |
| InternVL2.5-78B | 15.76% | 50.00% | 55.20% | 37.60% |
| MiniCPM-V-2.6-8B | 27.09% | 45.80% | 46.10% | 46.20% |
| Phi-3.5-vision | 10.84% | **54.20%** | **58.10%** | *Fail* |
| Ovis1.6-Gemma2-9B | 11.33% | 52.80% | 54.10% | 39.20% |
| Claude-3.7 | 22.66% | 55.20% | 48.80% | 43.40% |
| Untrained Human/Expert | 28% / 74% | 66% / 76% | 80% / 90% | 75% / 92% |

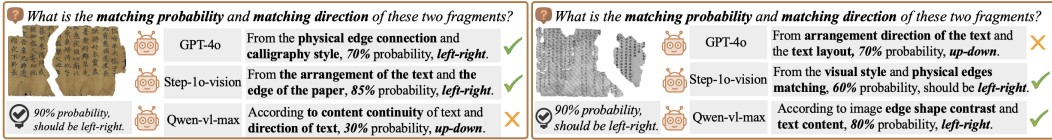

Figure 3: Qualitative showcases of Fragment Restoration results.

*Fragment Restoration* presents two persistent challenges in cross-image reasoning. *Scale Invariance Failure:* models achieve 47.64% accuracy on fragments of similar size, but collapse to 39.96% when paring large-to-small pieces (area ratio > 3:1). This suggests that LMMs may rely heavily on long-contour similarity and struggle with geometric reasoning, particularly when dealing with eroded or incomplete fragments. *Orientation Blindness:* while models achieve 52.18% in binary matching and 50.78% in probabilistic matching accuracy, their ability to predict spatial relationships is significantly lower (37.48% accuracy). Step-1o-vision-32k and SenseNova, which both achieved impressive restoration performance in binary and probabilistic matching, performed significantly worse in orientation inference, suggesting their over-reliance on texture patterns rather than contour and geometric semantics. We identify GPT-4o as the most reliable reference model, excelling in binary (59.04%), probabilistic (58.00%), and spatial orientation matching (72.00%), demonstrating the best alignment with ground truth in fragment positioning. Gemini and Qwen2.5-VL also exhibit strong consistency across all three question formats, making them promising alternatives. Among open-source LMMs, while overall performance lags 4.30% behind closed-source LMMs, Phi-Vision-3.5-4.2B achieved 58.10% accuracy in probabilistic matching, outperforming 67% of closed-source models.

We further explored that, as the number of input fragments increases, the pairwise restoration task transforms into a *combinatorial optimization* problem, where LMMs struggle to maintain accuracy due to the exponential increase in candidate pairs. Fragments that were previously correctly identified in pairwise settings are now frequently misclassified, indicating that the combinatorial complexity overwhelms current LLMs' capability. Integrating a small model for candidate filtering may reduce search space, and we leave for future exploration of building a fully autonomous restoring pipeline.

### 3.4 Cultural Study

To assess LMMs' multimodal alignment with cultural-historical priors, we design three progressively complex tasks: **Icon Recognition** evaluates foundational pattern recognition across 317 cultural motifs spanning religious icons, emblems and folk patterns. Each test case presents high-resolution artifact images with multi-choice questions, measured via **Accuracy**.

Table 5: Performance of LMMs on *Cultural Study*.

| | Icon Recognition Classification ↑ | Chronological Attribution (Accuracy) | | | | Artwork Caption BERTScore ↑ |
| --- | --- | --- | --- | --- | --- | --- |
| | | Classification ↑ | Binary ↑ | CoT ↑ | Retrieval ↑ | |
| GPT-4o | 92.78% | 26.45% | 38.80% | 35.14% | 63.77% | 62.16% |
| Step-1o-vision-32k | 83.44% | **61.23%** | **68.58%** | **54.35%** | **69.57%** | 62.01% |
| SenseNova | **96.86%** | *Fail* | 56.99% | *Fail* | 51.27% | 61.65% |
| Gemini-2.0-Pro-Exp | 96.00% | 40.58% | *Fail* | 38.04% | 56.88% | 60.63% |
| JT-VL-Chat | *N/A* | 31.88% | *N/A* | 31.88% | *N/A* | **63.69%** |
| Step-1V-32K | 11.22% | 38.04% | 55.74% | 39.86% | 48.91% | 61.80% |
| Qwen2.5-VL-3B | 31.67% | 32.97% | *Fail* | *Fail* | 40.22% | **63.40%** |
| Qwen2.5-VL-72B | **95.33%** | 32.97% | **55.19%** | 32.25% | 43.12% | 61.54% |
| Qwen2-VL-7B | 22.56% | 31.16% | 52.46% | 31.88% | 26.45% | 61.54% |
| InternVL2.5-2B | 3.22% | 33.70% | *Fail* | 44.20% | 19.93% | 63.02% |
| InternVL2.5-8B | 40.78% | **42.39%** | 48.63% | **44.57%** | 28.99% | 62.31% |
| InternVL2.5-78B | 94.78% | 33.70% | 50.00% | 33.70% | **47.10%** | 62.01% |
| MiniCPM-o-2.6-8B | 49.78% | 35.14% | 50.27% | 36.23% | 34.42% | 61.27% |
| Claude-3.7 | 95.56% | 31.88% | 51.37% | 42.03% | 49.28% | *N/A* |
| Untrained Human/Expert | 96% / 100% | 12% / 48% | 46% / 54% | *N/A* | *N/A* | *N/A* |

*Chronological Attribution*, challenges models to contextualize artifacts temporally. We also employ multi-format evaluation: Q&A, Multiple-choice, CoT (style feature verbalization before dating), Retrieval-augmented Q&A (comparative analysis with reference timelines).

*Artwork Caption*, assesses generative cultural narration through structured captioning of manuscript artwork and mural scenes. Beyond standard **BERTScore**, we introduce **manual expert scoring**, capturing the semantic accuracy, fluency, and contextual richness with historical images.

**Discussion.** *Icon Recognition* exposes significant gaps in LMMs' cultural-visual literacy. While closed-source LMMs achieve 82.93% accuracy (v.s. only 37.05% for open-source), we suspect this high performance suggests that some models may rely on pattern-based distinctiveness rather than genuine cross-modal understanding. Thus, we further restricted a two-image comparison against textual description. SenseNova exhibits consistent performance (96.86% on multiple-choice, 93.33% for binary picking, only a 3.53% drop), indicating robust cross-modal comprehension. In contrast, open-source LMMs such as Qwen2-VL-72B showed a sharp improvement, achieving 92.22% accuracy in the two-image selection setting, showed a sharp improvement, achieving 92.22% accuracy in two-image selection. This suggests that weaker cross-image reasoning previously hindered its performance, highlighting visual comparison in enhancing cultural recognition accuracy.

*Chronological Attribution* results indicate that generalized LMMs prioritize statistical prevalence over true historical reasoning, exhibiting a strong bias toward data-abundant eras (e.g., LMMs over-predict Tang Dynasty, which dominate the training data with the highest volume and quality of visual records). To mitigate this bias, we designed relative dating tasks ("Does artifact A predate B?") instead of requiring models to assign specific epochs, aligning with archaeological best practices. However, 45% of open-source models failed completely when random guessing was restricted, underscoring their failure in temporal reasoning. Step-1o-vision-32k peaked 69.57% through visual analogy, by matching unknown artifacts to reference examples.

For *Artwork Caption*, most LMMs produced generalized descriptions derived from pretrained common knowledge, rather than accurately identifying the specific story or historical event depicted. Thus, the BERTScore variations among models were minimal. To address this limitation, we incorporated manual expert scoring to evaluate semantic accuracy and contextual fidelity: GPT-4o and Step-1o emerged as the most aligned with human assessments (Step-1o could even name the image origin). These findings suggest that current LMMs lack cultural specificity and consistency in historical reasoning. While few-shot learning offers partial mitigation, the underlying misalignment between visual and cultural semantics hinders broader application requiring deep historical reasoning. We further introduced domain-informed dimensions (e.g., religious symbolism, facial expression, period, event specificity) to provide a more structured and interpretable evaluation of cultural understanding. Detailed results are included in Appendix L.

# 4 Conclusion and Outlook

We have presented **MS-Bench**, a large-scale, high-quality benchmark for visual-textual tasks in ancient manuscript analysis. Our hierarchical framework systematically addresses key challenges in manuscript research, to align with real-world archaeological and philological practices. Experiments reveal that current LMMs excel in structured, labor-intensive tasks (e.g., Handwritten Character Recognition, Allograph Normalization, and Calligraphy Style Classification). However, their performance in knowledge-intensive tasks remains constrained due to the lack of specialized training data and limited cultural grounding.

Looking forward, we aim to bridge these performance gaps through targeted domain adaptation and fine-tuning strategies, integrating extended datasets across diverse historical periods and manuscript styles. Incorporating smaller, specialized models could complement LMM capabilities. For instance, a dedicated visual perception module may better handle large aspect ratio manuscripts found in British collections. Moreover, given the combinatorial nature of multiple fragment matching, embedding a combinatorial solver alongside LMMs could offer a more structured and effective solution framework.

Further empirical studies involving domain experts and LMMs could yield valuable perspectives, accelerating the discovery of new manuscript connections and refining historical document analysis. With LMM-driven assistance, scholars can shift focus from repetitive manual tasks to higher-level historical analysis, facilitating deeper insights into new findings.

# 5 Acknowledgements

This work was in part supported by the National Natural Science Foundation of China (No. 62402429), National Key Research and Development Program of China (2024YFE0203700), "Pioneer" and "Leading Goose" R&D Program of Zhejiang (2025C02037), Young Elite Scientists Sponsorship Program by CAST (2024QNRC001), Ningbo Yongjiang Talent Introduction Programme (2023A-397-G), National Social Science Fund of China (No. 21BYY142, 14AZS001). The authors gratefully acknowledge the support of Zhejiang University Education Foundation Qizhen Scholar Foundation. The authors express their deep gratitude to Tiannan Zheng and Jingjing Shi for their invaluable advice and to Zhouyuan Li for his technical support.

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

# A    Related Work

The interdisciplinary efforts of applying vision technologies in assisting historical document studies have witnessed increasing attention in recent years. Previous research encompasses a diverse range of subjects [35], including Oracle Bones Inscriptions (OBI) [44], ancient manuscripts [23, 53, 54], frescoes and artifacts [50]. The associated tasks span traditional human scholarly efforts, covering palaeography (e.g., character recognition and textual analysis [25, 43, 46]), codicology (e.g., fragmented manuscript restoration and attribution [52]) and visual reconstruction [50]. OBI, a representative low-resource ideographic character, have emergent recent attention in deciphering unknown characters. The most recent effort from Guan et al. [22] proposed conditional diffusion models for first generate an initial decipherment from initial OBI input then refine with a style-referenced diffusion model for final OCR output. As for manuscript restoration, researchers typically adopted a Siamese-Neural Network by pairwise matching fragment candidates from a widespread type of materials including Oracle Bone [51], papyrus [30], manuscripts [6], dead sea scroll [31] and beyond. By capturing visual semantic information such as material, texture and font style, researchers achieved success in dating Arabic manuscripts [3], OBI [12] and Cuneiform Tablet [13]. Ancient relic restoration could be categorized into 2D restoration [18] and 3D reconstruction and repair [41]. Typical visual reconstructions vary from curve frescoes repair, character in-painting [45] to 3D fragment reconstruction [33].

We also acknowledged that some early-stage LLM and LMM applications in ancient manuscript and cultural heritage processing have emerged recently. [54] pioneered in introducing LMM for small manuscript pieces restoration in a grid like pattern, however they failed to further delve into broader LMM candidates and question type influence on matching performance. [11] also proposed a LLM focusing on various downstream task for ancient Chinese text understanding, including translation, word explanation and beyond. [49] highlighted its capability to recognize characters from bamboo slips in its performance demonstration, showcasing the growing interest in expanding LMM applications for end-to-end ancient artifact analysis.

The most relevant work from [15] proposed the first benchmark tailor for LMM on OBI processing, encompassing 5 domains and 4 types of questions (what, why, how, where). The key differences between OBI-Bench and our work mostly lie in *research focus and data source*: 1) *Scope of Study:* OBI-Bench primarily focuses on Oracle Bone, with 80% of its tasks dedicated to character identification. In contrast, our MS-Bench encompasses a broader range of tasks, including text recognition for ancient character transcription, textual analysis for symbols and style classification, materiality studies focusing on puzzle-like matching (which is a key challenge for manuscript study) and iconographic analysis. 2) *Data Source Difference:* benefit from OBI's earlier research foundation on character recognition, OBI-Bench primarily integrates and supplements existing public datasets. In contrast, our benchmark data is mostly curated from sketch with expert guidance. 3) *Prompt Engineering Strategies:* We further explore the impact of different prompting strategies, specifically CoT and Visual Retrieval Augmentation (current cost-efficient solution given the scarcity of domain data), and compare their effects on LMMs' performance on both positive and negative sides.

# B  Dataset Details

Here we given more implementation details of our benchmark dataset curation, we elaborate the data source, preprocessing, annotation, image and data quantity. We performed *scale normalization* for all images to ensure a consistent scale around $600 \times 600$ pixel, to prevent issues caused by excessively large or small manuscript fragments. We performed *color normalization* for images sourced from global museum.

For *Handwritten Character Recognition*, data source from our collaborated Dunhuang manuscript research team. We filtered out scrolls with an aspect ratio exceeding 3:1, removing overly long manuscripts that could distort model evaluation. We applied RushiOCR for initial machine transcription, followed by manual double checking to filter out incorrect characters on fragmented image edges. For *Allograph Normalization* and *Writing Symbol Detection*, we implemented super-resolution since most cropped image have relatively smaller sizes. These tasks use data source from expert manual collections with human-annotated golden labels.

For *Calligraphy Style Classification*, half of our data is sourced from publications with ground truth while the rest source from open-source museum repositories. We uniformly process these images to grayscale.

For *Damage Assessment*, our data is sourced from open museum repositories, with its human-evaluated damage severity serving as ground truth. No further implementation is required.

For *Manuscript Restoration*, we applied data source from expert collection. We removed fragment background and transparent alpha channel. All images are transformed to grayscale, because there exists matching pairs from two distinct museums of both color and grayscale digitized image. We aim to prevent color from introducing additional biases in the model's decision-making. Since real-world ground truth is scarce, we introduced a simulated shredding method for pairwise matching data augmentation.

For the other tasks, our data is sourced from publications, books and research papers. We cropped these images, extracted text descriptions and labels manually, followed by minimal adjustments (resize, color normalization, background removal) to ensure consistent image quality.

Here we provide more detailed analyses of manuscript dynasties, writing styles (fonts), image sources, manuscript status, and the distribution of writing symbols in Table 6. Some statistics are not available for every image in our dataset. For example, the dating of certain manuscripts used in the Material Study task remains under debate among archaeologists, so for consistency and accuracy, we only include manuscripts used in *Chronological Attribution* tasks when reporting dynasty-level statistics. We also acknowledge an imbalance in calligraphic style distribution, this reflects the real-world composition of the Dunhuang manuscripts, in which over 90% of manuscripts are Buddhist scriptures written in Regular Script.

| Dimension | Statistics |
|---|---|
| **Image Source** | 54.64% Specialist Collection, 24.99% Publications & Research Papers, 12.81% Museum, 7.56% Others |
| **Writing Symbol** | 48.46% Deletion, 19.90% Repetition, 18.28% Hierarchy, 6.71% Pause, 6.63% Tick |
| **Calligraphy Style** | 91.38% Regular Script, 2.87% Running Script, 2.87% Clerical Script, 2.87% Cursive Script |
| **Dynasty** | 33.33% Tang Dynasty, 33.33% Wei-Jin Dynasty, 16.67% Five Dynasties, 16.67% Yuan Dynasty |
| **Manuscript Status** | 64.91% Damaged, 35.09% Intact |

Table 6: Detail dataset statistics.

## C  Experiment Details

### C.1  Model Selection

We select LMM candidates based on their overall performance on mainstream LMM benchmarks, according to their performance on OpenVLM Leaderboard [26]. Ranking by *Average Score* on leaderboard, we included `Step-1o-vision-32k` [36], `SenseNova` [34], `GLM-4V-Plus-0111` [21], `HunYuan-Standard-Vision` [40], `Qwen2.5-VL` Series [9], `InternVL-2.5` Series [14], `Qwen2-VL` Series [42], `Qwen-VL-Max` [8], `Gemini` Seires [39], `JT-VL-Chat` [16], `Step-1V-32K` [37], `GPT-4o` [24], `Ovis1.6-Gemma2-9B` [29], `Claude-3.5-Sonnet-20240122` [4], `MiniCPM` Series [49], `Claude-3.7` [5], `Valley-Eagle-7B` [48], `DeepSeek-VL` Series [28, 47], `LLaVA-OneVision-7B` [27], `GPT-4V` [2]. By the time we perform experiments, the top-tier LMMs such as `TeleMM`, `BailingMM` Series and `Taiyi` have no public access (either open-source model or API reference), `Phi-3.5-vision` Series [1] is not on the leaderboard. We do not include extensive comparisons of LMMs' backbones and parameter sizes in this paper, since they are more intuitively organized on leaderboards.

When selecting candidate LMMs, we took the following points into consideration:

1) *Prioritizing Strong General-Purpose Models:* we provide direct contrast between general multi-modal proficiency and domain-specific performance in manuscript tasks.

2) *Various Open-Source Model Sizes:* open-source models were primarily selected within the 7-10B parameter range, as they are the most commonly used configurations for individual deployment. Additionally, we included a limited number of larger models (>70B) and smaller device LMMs (<3B) to evaluate performance in both resource-rich and resource-constrained scenarios.

3) *Practical Usability:* in some manuscript research tasks (e.g., allograph normalization and calligraphy style classification), scholars may prioritize models with easier access, resource-efficient (with API references), and deployable on edge devices, rather than models requiring complex setups.

## D  Implementation

To maintain consistence performances and practical usability, we prioritize LMMs' offical APIs for experiments if available. The models for API implementation include all closed-source LMMs and all `Qwen2.5-VL` and `Qwen2-VL` series models. We deploy other LMMs on a server equipped with AMD Processors and NVIDIA A100-PCIE-40GB GPUs. To maintain controllable comparisons with open-source LMMs' claimed performance, our deployment and inference were performed under each LMMs' standard recommendation in their respective official repository. We set deterministic parameters when inference ( `temperature = 0`, `top-k = 1`) to avoid randomness. For those API references that we could not explicitly avoid randomness (e.g., GPT-4o APIs would response slightly differently even with deterministic parameters), we performed two-round experiments and take the average performance. No LMMs observed significant variation between different rounds of experiments.

We performed minimal adjustments on our task prompt as preliminary experiments (tested with a mini-batch of around 30 queries each task) without influencing task design. This allows each LMM to perform at its optimal capability, providing a fair and comprehensive evaluation.

## E  Extended Discussion

We acknowledged that there exists differences in instruction following capabilities across LMMs, smaller models (<10B) tend to get confused with our task instructions more frequently. `GPT-4o`, `Gemini` series, `GLM` series, `InternVL` series, `Valley-Eagle-7B` and `Qwen-VL-Max` show satisfying instruction understanding. `QVQ`, `DeepSeek-VL` series and `MiniCPM` series demonstrate notable decline in adhering to user prompts, frequently failing to comprehend the requirements of our complex task instructions.

In MS-Bench, closed-source LMMs developed by Chinese researchers outperformed others by an average of 11.73% in *Textual Recognition* tasks. Apart from models like `MiniCPM`, which explicitly

emphasize ancient character recognition proficiency, we hypothesize that OCR training data selection may be influenced by the cultural background of dataset curators or researchers.

However, we did not observe significant gap in other multimodal tasks, where performance is more tightly linked to visual perception capabilities rather than purely cultural knowledge generation. Notably, inherent visual-textual alignment capabilities play a more decisive role in overall model performance, as evidenced by strong baselines of `Gemini` and `GPT-4o`. This cultural bias diminishes as model size increases and with continued iterations, suggesting that scaling and ongoing model refinement help mitigate pretraining-induced cultural imbalances.

## F    Extended Discussion on Prompting Strategies across Task Types

Prompt design plays a critical role in the performance of LMMs across different types of manuscript analysis tasks. We present here an extended discussion of various prompting strategies.

In general, **CoT is more effective when task benefits from explicit step-wise reasoning and disambiguation.** For example, in the *Writing Symbol Detection* task, CoT helps models correctly disambiguate visual marks by guiding them to focus on regions outside the main text area. This helps distinguish between small marks like "Tick Symbol" and a larger similar shaped character or strokes. However, **CoT can harm performance in perception-heavy classification tasks that rely on holistic visual judgment or stylistic intuition**. In tasks like *Calligraphy Style Classification* or *Chronological Attribution*, we observed that CoT often led to over-analysis or model hesitation. For instance, models like `Gemini` or `InternVL` under CoT prompts sometimes defaulted to vague or generic answers such as "Regular Script" or "I am not sure", due to a misalignment between expert-designed complex reasoning logic and model's internal visual judgment. It is inherently difficult to "teach" models to differentiate subtle holistic patterns through textual descriptions alone (e.g., between Clerical and Running Script), especially without pretraining on such domain-specific stylistic variations.

In contrast, **V-RAG offers more stable and interpretable improvements by grounding model predictions in explicit visual comparisons with annotated reference exemplars**, reducing reliance on free-form textual reasoning. For example, in *Writing Symbol Detection*, model directly "compares" given manuscript with reference samples to locate matching patterns, benefiting from its inherent visual grounding capability without exterior domain knowledge required. In *Chronological Attribution*, for example, both "Tang Dynasty" and "Wei-Jin Dynasty" figures may exhibit richer color palettes, but "Tang" figures are characteristically rounder in facial and body structure. V-RAG allows models to align unknown inputs with these reference traits, improving relative dating accuracy without relying solely on unstable textual reasoning. V-RAG thus excels in situations where cultural distinctions are visually encoded but hard to verbalize.

To evaluate prompt sensitivity, we conducted an ablation comparing simple **role-play** prompting and **few-shot prompting with reference examples** on the *Calligraphy Style Classification* task. The results in Table 7 show model-dependent effects:

| Model | Role-Play | Few-Shot |
|---|---|---|
| GPT-4o | 0.6767 | 0.7467 |
| Step-1o-vision-32k | 0.7067 | 0.7833 |
| Qwen-VL-Max | 0.7700 | 0.7533 |
| Qwen2.5-VL-72B | 0.7867 | 0.8033 |

Table 7: Performance comparison of prompting strategies on *Calligraphy Style Classification*.

Based on these results and previous discussions, we offer the following recommendations for researchers future benchmark builders:

1. Role-playing is a simple yet effective strategy that improves task alignment.

2. Provide explicit reasoning steps or focal aspects often works better than relying on the model's own decomposition.

3. Clarify task-specific terminology helps reduce ambiguity.

4. Incorporate multiple prompting strategies, and evaluate not only absolute performance but also interactions between prompts, task type, and model characteristics (e.g., hallucination resistance, visual grounding);

5. For culturally grounded tasks, consider co-designing prompts with domain experts and iteratively refining them.

| | Handwritten Character Recognition | | Allograph Normalization (Accuracy) | | |
| --- | --- | --- | --- | --- | --- |
| | Accuracy ↑ | Edit Distance ↓ | Easy ↑ | Medium ↑ | Hard ↑ |
| Human (Untrained) | *Fail* | *Fail* | 84% | 72% | 48% |
| Human (Expert) | 91% | 19.65 | 96% | 96% | 84% |
| GPT-4V | 6.77% | 415.01 | *Fail* | *Fail* | *Fail* |
| GPT-4o | *Fail* | *Fail* | 65.41% | 39.09% | 22.54% |
| Step-1o-vision-32k | 80.19% | 125.39 | **89.04%** | **77.16%** | **58.10%** |
| SenseNova | **82.88%** | 113.30 | 88.70% | 74.62% | 56.69% |
| JT-VL-Chat | *Fail* | *Fail* | 29.79% | 11.93% | 7.75% |
| Gemini-2.0-Flash | 74.80% | **60.05** | 71.23% | 47.21% | 36.97% |
| Gemini-1.5-Pro | 72.58% | 97.83 | 39.04% | 24.11% | 17.61% |
| Claude-3.5-Sonnet | 48.57% | 128.85 | 50.68% | 29.19% | 20.42% |
| GLM-4V-Plus | 81.31% | 81.62 | 74.66% | 52.79% | 50.00% |
| Qwen-VL-Max | 59.23% | 202.92 | 71.00% | 49.00% | 35.21% |
| Hunyuan-Vision | 76.10% | 112.74 | 64.04% | 41.37% | 36.27% |
| Hunyuan-Turbo-Vision | 76.91% | 128.08 | 66.10% | 50.51% | 40.49% |
| Step-1V-32K | 57.06% | 224.36 | 60.62% | 38.32% | 32.04% |
| Gemini-2.0-Pro-Exp | 76.53% | 78.54 | 61.64% | 43.91% | 33.80% |
| Qwen2-VL-2B | 28.40% | 325.51 | 77.74% | 56.85% | 38.73% |
| Qwen2-VL-7B | 46.40% | 425.25 | 44.86% | 29.70% | 25.00% |
| Qwen2-VL-72B | 51.78% | 236.80 | 30.14% | 16.75% | 11.27% |
| Deepseek-VL2 | *Fail* | *Fail* | 37.33% | 20.81% | 11.97% |
| Deepseek-VL | *Fail* | *Fail* | *Fail* | *Fail* | *Fail* |
| InternVL2.5-2B | 40.93% | 5444.89 | 62.33% | 49.49% | 38.73% |
| InternVL2.5-8B | 43.15% | 2311.89 | 56.51% | 48.73% | 34.86% |
| InternVL2.5-78B | *N/A* | *N/A* | **90.41%** | **81.98%** | **63.73%** |
| GLM-4V-9B | 69.16% | 299.94 | 63.36% | 41.62% | 33.10% |
| LLaVA-OneVision-7B | *Fail* | *Fail* | *Fail* | *Fail* | *Fail* |
| MiniCPM-V-2.6-8B | 45.42% | 920.30 | 53.77% | 24.11% | 15.14% |
| Phi-3.5-vision | *Fail* | *Fail* | *Fail* | *Fail* | *Fail* |
| Ovis1.6-Gemma2-9B | *Fail* | *Fail* | *Fail* | *Fail* | *Fail* |
| Valley-Eagle-7B | 34.96% | 581.91 | 80.48% | 58.83% | 40.85% |
| MiniCPM-o-2.6-8B | 46.90% | 673.01 | 33.73% | 15.74% | 6.34% |
| Qwen2.5-VL-3B | 64.69% | 456.16 | 52.05% | 36.04% | 27.11% |
| Qwen2.5-VL-7B | **72.99%** | **92.63** | 61.30% | 42.39% | 29.23% |
| Qwen2.5-VL-72B | 69.39% | 102.99 | 55.82% | 39.85% | 26.76% |
| Claude-3.7 | *Fail* | *Fail* | 53.77% | 34.77% | 21.83% |
| QVQ-72B | 46.90% | 673.01 | 32.53% | 23.60% | 17.61% |

Table 8: Overall performance of LMMs on ***Textual Recognition*** tasks. The best LMM of each set is **bold**, the second-best is underlined. "*N/A*" indicates such LMM does not accept multiple visual inputs. "*Fail*" indicates such LMM could not solve the task (simply refuse to answer, all return same answers or generate random guesses). These special notations are maintained in the following tables.

# G Extended Handwritten Character Recognition Discussion

## G.1 LLM Performance

Our experimental results in Table 8 confirm that LMM OCR performance adhere to scaling law, where larger LMMs outperform smaller ones. Notably, LMMs less than 10B parameters struggle to independently complete OCR tasks for ancient handwritten manuscripts. Common failure cases include:

1) *Inaccurate text localization:* ancient manuscripts are typically vertically oriented, written from right to left, which differs from modern reading conventions that dominant in pretraining data distribution. Even explicitly specified in system and user prompt, open-source smaller LMMs frequently disregard these instructions (poor instruction following), defaulting to left-to-right output. This is a primary reason that certain LMMs exhibit abnormally high Edit Distance.

2) *Edit Distance and Accuracy not fully correlated:* while lower Edit Distance indicates stronger visual grounding (i.e., better text localization), higher Accuracy reflects superior character-level recognition performance. Discrepancies between these two metrics suggest that some LMMs excel at detecting text regions but struggle with precise transcription (e.g., *Claude-3.5-Sonnet*), whereas others achieve high OCR accuracy but misaligned text placements (e.g., `MiniCPM-V-2.6-8B`).

3) *Repetitive output and looping errors* more frequently appear in this task, because some manuscripts (particularly Buddhist scriptures) naturally contain poetic structures with repeating phrases. Smaller LMMs frequently fall into output loops (e.g., *InterVL2.5-8B*), repeatedly generating the same text segments.

4) *Fragmented manuscripts has more incomplete characters* which lead to poor OCR accuracy at manuscript broken edges and torn contours. This is another major factor contributing to poor OCR performance besides model inability.

In contrast to traditional OCR models that single-character recognition failure lead to substantial inaccuracies, end-to-end LMM-based OCR ensures that detected text aligns semantically. However, this probabilistic text generation approach introduce an inherent trade-off: LMMs favor common vocabulary over precise text transcription. This automated "correction" based on learned linguistic patterns, often lead to merging or reordering content according to pretraining biases. This sampling-based generation mechanism also explains why LMMs perform well on allograph normalization for script variation: they map unfamiliar characters to more common ones, which is the allograph normalization target.

# H Extended Allograph Normalization Discussion

## H.1 Human View

Allographs are not spelling errors but variant forms of the same character. The emergence of allographs is due to the following reasons: *Differences in writing habits and literacy levels among writers*, leading to missing or adding strokes. *Simplification of complex characters:* difficult-to-write characters are replaced with a simpler alternative with the same pronounce, also known as *Phonetic Loan Characters. Evolution of characters over time* due to historical and linguistic changes. *Imperial naming taboos*, which lead to modifications in character radicals. Since Chinese is a meaning-based writing system, the visual representation of characters has continuously evolved following two directions. "More Complex": adding radicals to create more distinct and precise characters due to limited number of existing symbols. "Simpler": reducing strokes for ease of writing, often influenced by popular or informal script styles.

Given the semantic nature of Chinese characters, allographs can be generally categorize into 5 types: 1) *Script Variation:* changes in stroke addition or omission. 2) *Radical Substitution:* differences in the semantic component which lead to totally different meanings. 3) *Structural Transformation:* variation in character formation, such as differences in traditional and simplified Chinese. 4) *Radical Repositioning:* the radicals shift between left-right to top-bottom structure and beyond. 5) *Semantic Component Variation:* different radicals conveying the same meaning, resulting in multiple character forms.

## H.2  LMM Performance

We constructed random re-sampling among LMM outputs and statistical analysis experimental results, as in Figure 8 indicates `SenseNova` and `Step-1o-vision-32k` consistently outperformed other models. Specifically, `Step-1o-vision-32k` demonstrates a clear advantage on *Script Variation*, *Structural Transformation* and *Radical Repositioning*, aligning with its overall superior performance across the dataset.

Among all allograph types, *Script Variation* accounts for the largest amount of data and exhibits the highest overall accuracy. In contrast, LMMs performed significantly worse on *Structural Transformation* and *Radical Substitution*, as these involves substantial changes in character composition and visual spacial distribution, making recognition extremely challenging. This trend aligns with human performance, meaning LMMs perform allograph normalization based on visual perceptual inference, similar to manual visual intuition.

We also noticed that our dataset categorization is based on *frequency* rather than *difficulty or complexity*. LMM performance should be relatively uniformed across all three frequency levels. However, results show otherwise, indicating LMM performance is heavily influenced by pretraining data biases. This aligns with real-world data distributions, where the most frequent allograph type exists more frequently in pre-training dataset, and consequently achieves the highest accuracy. While rarer characters and less common variation types are normally dismissed in pretraining, thus suffering from lower analogy-based inference accuracy.

## H.3  Model Scale Anomaly Discussion

We observed an anomaly where a smaller LMM outperformed a larger one, particularly within the *Script Variation* subtype. Based on our analysis, this performance reversal stems from allograph type-specific sensitivities:

In *Script Variation* type, which involves subtle stroke-level differences, smaller models sometimes perform better likely due to their reduced visual sensitivity, which makes them less susceptible to insignificant changes. In contrast, the larger models perform better on semantic component variation cases, where deeper reasoning about component-meaning alignment is needed. We propose that collaborative inference between large and small models may combine their complementary strengths to improve overall accuracy.

For the more challenging *Radical Repositioning* cases, where both models struggle due to substantial allograph deviation, we believe that domain-specific continued pretraining is needed. Given training efficiency constraints, smaller models may be more practical for targeted fine-tuning.

We therefore view this as a design opportunity, where future systems can combine the efficiency of smaller models with the stronger overall performance of larger models to achieve more robust and adaptable solutions. Future research may include model collaboration strategies, selective decoding, or fine-tuning smaller models under expert supervision.

## H.4  Future Direction

In real-world manuscript research, allograph recognition is often coupled with dating analysis and scribe attribution, as specific allograph styles can assist in indicating particular historical periods and individuals. Given this strong temporal correlation, we advocate for future work to incorporate a time dimension into allograph analysis. By further pretraining and instruction tuning, LMMs should be enhanced to leverage visual features for manuscript dating, enabling a more academically valuable task.

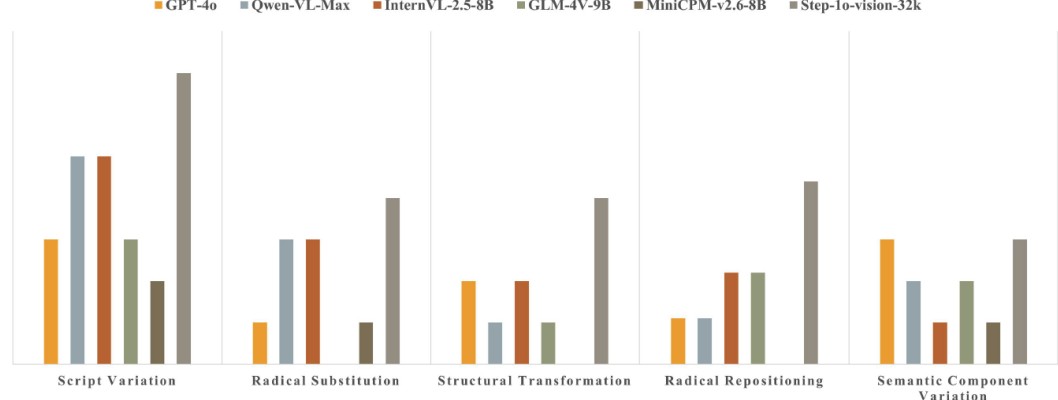

Figure 4: Detail statistics of LMM performance across different types of allograph. We randomly sample a subset from all allographs and categorize by their variation types.

| Allograph | Ground Truth | GPT-4o | Qwen-VL-Max | InternVL-2.5-8B | GLM-4V-9B | MiniCPM-v2.6-8B | Step-1o-vision-32k |
|---|---|---|---|---|---|---|---|
| 庱 | 底 | 广❌ | 底✅ | 底✅ | 底✅ | 石❌ | 底✅ |
| 幹 | 幹(干) | 敏❌ | 幹✅ | 干✅ | 龄❌ | 伥（長）❌ | 铃❌ |
| 壏 | 鹽(盐) | 盐✅ | 鹽✅ | 塩✅ | 盘❌ | －❌ | 盐✅ |
| 熱 | 熱(热) | 熟❌ | 熟❌ | 焚❌ | 熬❌ | 就❌ | 热✅ |
| 啓 | 啟(启) | 欲❌ | 唠❌ | 启✅ | 弭❌ | 恐❌ | 启✅ |
| 刪 | 删 | 刑❌ | 删✅ | 册❌ | 删✅ | 帝❌ | 删✅ |
| 髻 | 髻 | 語❌ | 善❌ | 髻❌ | 赔❌ | 華❌ | 髻✅ |
| 盛 | 盛 | 岁❌ | 感❌ | 盛✅ | 咸❌ | 城❌ | 盛✅ |
| 職 | 職(职) | 职✅ | 躲❌ | 軾❌ | 戧❌ | 载❌ | 职✅ |
| 憙 | 喜 | 意❌ | 喜✅ | 意❌ | 熹❌ | 惠❌ | 熹❌ |

Figure 5: Cases of LMM results with *Allograph Recognition* tasks, categorized by allograph types.

| | Writing Symbol Detection (Accuracy) | | | | Calligraphy Style Classification (Accuracy) | | |
| --- | --- | --- | --- | --- | --- | --- | --- |
| | Q&A ↑ | Binary ↑ | CoT ↑ | Retrieval ↑ | Q&A ↑ | CoT ↑ | Retrieval ↑ |
| Human (Untrained) | 36% | 52% | *N/A* | *N/A* | 40% | *N/A* | *N/A* |
| Human (Expert) | 88% | 98% | *N/A* | *N/A* | 78% | *N/A* | *N/A* |
| GPT-4V | 10.51% | 65.83% | 17.32% | 35.21% | 37.25% | 42.75% | 54.00% |
| GPT-4o | 11.48% | 62.38% | 12.84% | 24.12% | 37.25% | 40.75% | 54.75% |
| Step-1o-vision-32k | 24.90% | **69.29%** | 26.85% | **40.94%** | **60.75%** | **63.50%** | **66.25%** |
| SenseNova | 13.23% | 63.53% | 9.73% | 5.64% | 53.44% | 52.42% | 24.94% |
| JT-VL-Chat | 8.95% | 51.06% | 4.86% | *N/A* | 34.25% | 33.75% | *N/A* |
| Gemini-2.0-Flash | 6.72% | 55.09% | 20.55% | 34.98% | 53.00% | 56.75% | 58.25% |
| Gemini-1.5-Pro | **29.05%** | 68.91% | 38.93% | 36.17% | 48.25% | 47.50% | 45.75% |
| Claude-3.5-Sonnet | 8.45% | 48.94% | 10.28% | 34.98% | 50.50% | 54.25% | 64.50% |
| GLM-4V-Plus | 0.97% | 51.06% | 0.39% | *N/A* | 44.86% | 39.85% | 43.11% |
| Qwen-VL-Max | 12.28% | 13.32% | 7.20% | 25.88% | 43.58% | 44.08% | 49.62% |
| Hunyuan-Vision | 3.95% | 33.78% | 9.49% | 10.33% | 47.75% | 50.00% | 27.00% |
| Hunyuan-Turbo-Vision | 1.38% | 31.67% | 8.89% | 5.34% | 57.25% | 49.75% | 60.00% |
| Step-1V-32K | 11.48% | 64.49% | 12.84% | 25.88% | 59.25% | 62.00% | 61.25% |
| Gemini-2.0-Pro-Exp | 22.73% | 67.95% | **40.12%** | 37.55% | 43.00% | 40.50% | 63.25% |
| Qwen2-VL-2B | *Fail* | 38.00% | *Fail* | 4.67% | *Fail* | 24.94% | 24.69% |
| Qwen2-VL-7B | 11.07% | 46.64% | 10.87% | 15.02% | 27.00% | 28.00% | 26.00% |
| Qwen2-VL-72B | 12.65% | 63.53% | 13.24% | 25.89% | 44.50% | 43.25% | 53.50% |
| Deepseek-VL2 | 3.16% | 48.56% | 18.18% | 17.59% | 27.75% | 31.00% | 43.50% |
| Deepseek-VL | 0.78% | 46.64% | 5.84% | 16.34% | *Fail* | 16.50% | *Fail* |
| InternVL2.5-2B | 4.09% | 51.06% | 13.23% | 5.06% | 21.25% | *Fail* | 20.75% |
| InternVL2.5-8B | 2.72% | 46.45% | 19.26% | 20.23% | 30.25% | 31.00% | 26.00% |
| InternVL2.5-78B | 4.09% | 52.21% | *Fail* | 32.49% | 40.50% | 38.25% | 24.75% |
| GLM-4V-9B | 6.23% | 50.86% | 1.17% | *N/A* | 46.50% | 49.75% | *N/A* |
| LLaVA-OneVision-7B | *Fail* | **70.44%** | 14.40% | 18.48% | 26.75% | 29.25% | 24.75% |
| MiniCPM-V-2.6-8B | 9.73% | 45.49% | 7.20% | 15.76% | 41.75% | 41.75% | 44.25% |
| Phi-3.5-vision | 0.19% | 68.33% | *Fail* | **39.30%** | 25.25% | 24.75% | 25.00% |
| Ovis1.6-Gemma2-9B | 2.14% | 38.20% | 2.72% | 8.95% | 39.00% | 35.75% | 32.00% |
| Valley-Eagle-7B | 5.64% | 61.42% | 13.81% | 11.09% | 30.25% | 38.25% | 25.25% |
| MiniCPM-o-2.6-8B | 12.65% | 47.60% | 7.20% | 21.21% | 40.50% | 42.75% | 50.50% |
| Qwen2.5-VL-3B | **31.82%** | 39.35% | **22.13%** | 13.24% | 32.25% | 31.50% | 25.00% |
| Qwen2.5-VL-7B | 6.13% | 41.46% | 9.49% | 14.43% | 49.25% | 49.25% | 55.50% |
| Qwen2.5-VL-72B | *Fail* | 67.37% | 8.30% | 21.15% | **60.25%** | **59.25%** | **63.00%** |
| Claude-3.7 | 3.84% | 50.67% | 7.49% | 29.17% | 58.50% | 61.00% | 66.00% |
| QVQ-72B | 1.78% | 59.12% | 7.51% | 20.55% | 48.25% | 41.75% | 36.25% |

Table 9: Overall performance of LMMs on *Textual Analysis* tasks. The best LMM of each set is **bold**, the second-best is underlined.

# I Extended Writing Symbol Detection Discussion

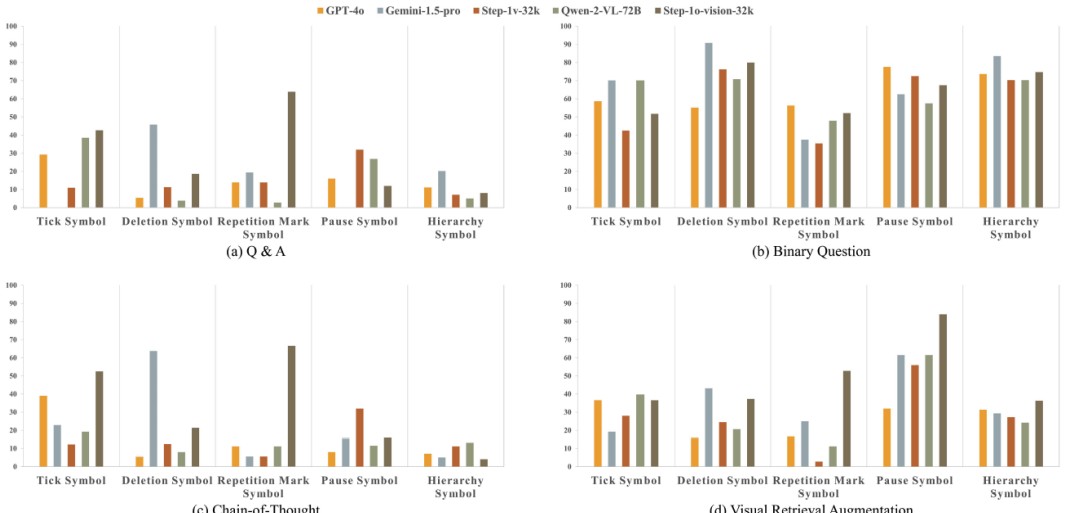

Figure 6: Statistics of LMM performance across different types of writing symbol and various question formats. There exists easier symbols with high accuracy and harder symbols to recognize. LMM also demonstrate their preferences on different symbol types.

## I.1 LMM Performance

We conducted grouped sampling and statistical analysis based on symbol types within the dataset. This task requires LMMs to visually locate symbols based on textual descriptions, making precise text-visual alignment critical.

Our results confirm that CoT reasoning benefits this task which require fine-grained visual recognition. With CoT, LMMs actively attempt to understand the meaning and shape of symbols, paying attention to distinct textual markers rather than treating them as mere visual noise. This aligns with our original motivation for incorporating CoT, as its structured reasoning process reinforces text-visual alignment and slows down judgment making.

We also observed that certain symbols are inherently easier to recognize. For example, *Pause Symbols* are more distinguishable due to their clear visual contrast against surrounding text. Similarly, symbols with distinct colors (e.g., red annotations) are more visually salient among black characters.

Interestingly, under the same prompt and visual input conditions, different models exhibited symbol-specific preferences and recognition biases. `Gemini` excelled in recognizing *Deletion Symbol*, while *Step-1o-Vision-32k* preferred *Repetition Symbol* (a symbol which other LMMs struggle with).

## I.2 Future Direction

Given that manual completion of symbol detection is highly labor-intensive, multi-model collaboration or enhancing prompts with additional prior knowledge (e.g., description of symbol position or semantic meanings) may be necessary. At present, Visual Retrieval Augmentation remains the most effective solution.

| | Damage Assessment(Accuracy) | Fragment Restoration (Accuracy) | | |
| --- | --- | --- | --- | --- |
| | Multiple Choice ↑ | Binary ↑ | Probabilistic ↑ | Orientation ↑ |
| Human (Untrained) | 28% | 66% | 80% | 75% |
| Human (Expert) | 74% | 76% | 90% | 92% |
| GPT-4V | *Fail* | 64.17% | 55.54% | 47.84% |
| GPT-4o | 11.82% | 62.76% | 62.01% | **79.01%** |
| Step-1o-vision-32k | 19.31% | 68.38% | **64.80%** | *Fail* |
| SenseNova | 20.81% | **70.09%** | 63.26% | 7.19% |
| JT-VL-Chat | 13.30% | *N/A* | *N/A* | *N/A* |
| Gemini-2.0-Flash | **43.35%** | 41.45% | 45.03% | 42.59% |
| Gemini-1.5-Pro | 16.26% | 47.66% | 42.63% | 47.84% |
| Claude-3.5-Sonnet | 25.62% | 56.79% | 41.48% | 35.49% |
| GLM-4V-Plus | 23.65% | 59.79% | 56.45% | 31.79% |
| Qwen-VL-Max | *Fail* | 61.48% | 57.60% | 55.56% |
| Hunyuan-Vision | 31.03% | 41.33% | 40.69% | 46.60% |
| Hunyuan-Turbo-Vision | 37.93% | 41.33% | 42.74% | 44.75% |
| Step-1V-32K | 28.71% | 59.13% | 58.97% | 49.07% |
| Gemini-2.0-Pro-Exp | 40.89% | *Fail* | 43.20% | 45.37% |
| Qwen2-VL-2B | 13.30% | 44.85% | 42.63% | *Fail* |
| Qwen2-VL-7B | 26.11% | 51.87% | 52.80% | 32.10% |
| Qwen2-VL-72B | 33.50% | 47.42% | 51.54% | 31.17% |
| Deepseek-VL2 | 30.54% | 49.30% | 48.80% | 20.99% |
| Deepseek-VL | 18.23% | 47.10% | 42.86% | *Fail* |
| InternVL2.5-2B | 21.18% | **58.78%** | 36.91% | 17.59% |
| InternVL2.5-8B | 15.27% | 40.63% | 43.66% | 30.86% |
| InternVL2.5-78B | 15.76% | 58.55% | 50.17% | 38.27% |
| GLM-4V-9B | 43.35% | *N/A* | *N/A* | *N/A* |
| LLaVA-OneVision-7B | 16.75% | 41.69% | 44.34% | *Fail* |
| MiniCPM-V-2.6-8B | 27.09% | 40.40% | 43.66% | 39.51% |
| Phi-3.5-vision | 10.84% | 46.49% | **53.37%** | *Fail* |
| Ovis1.6-Gemma2-9B | 11.33% | 44.50% | 52.91% | 40.74% |
| Valley-Eagle-7B | 18.23% | *Fail* | 44.34% | 48.46% |
| MiniCPM-o-2.6-8B | 19.31% | 41.45% | 37.94% | 32.41% |
| Qwen2.5-VL-3B | 15.76% | 51.17% | 42.97% | *Fail* |
| Qwen2.5-VL-7B | **43.84%** | *Fail* | 45.14% | **50.62%** |
| Qwen2.5-VL-72B | 12.81% | **58.78%** | 42.86% | 44.44% |
| Claude-3.7 | 22.66% | 60.54% | 42.40% | 41.05% |
| QVQ-72B | 37.44% | 42.62% | 40.46% | 50.00% |

Table 10: Overall performance of LMMs on *Materiality Study* tasks. The best LMM of each set is **bold**, the second-best is underlined.

| | Icon Recognition | Chronological Attribution (Accuracy) | | | | Artwork Caption (Score) |
| | Classification ↑ | Classification ↑ | Binary ↑ | CoT ↑ | Retrieval ↑ | BERTScore ↑ |
|---|---|---|---|---|---|---|
| Human (Untrained) | 96% | 12% | 46% | *N/A* | *N/A* | *N/A* |
| Human (Expert) | 100% | 48% | 54% | *N/A* | *N/A* | *N/A* |
| GPT-4V | 90.67% | *Fail* | *Fail* | *Fail* | 42.03% | 61.05% |
| GPT-4o | 92.78% | 26.45% | 38.80% | 35.14% | 63.77% | 62.16% |
| Step-1o-vision-32k | 23.33% | **61.23%** | **68.58%** | **54.35%** | **69.57%** | 62.01% |
| SenseNova | **96.86%** | *Fail* | 56.99% | *Fail* | 51.27% | 61.65% |
| JT-VL-Chat | *N/A* | 31.88% | *N/A* | 31.88% | *N/A* | **63.69%** |
| Gemini-2.0-Flash | 95.78% | 36.23% | 52.46% | 38.41% | 52.54% | 61.95% |
| Gemini-1.5-Pro | 94.56% | 23.19% | 50.00% | 19.20% | 48.19% | 62.04% |
| Claude-3.5-Sonnet | 94.11% | 35.87% | 51.09% | 39.49% | 46.38% | 61.48% |
| GLM-4V-Plus | 92.56% | 33.70% | 48.63% | 34.78% | 44.20% | 61.91% |
| Qwen-VL-Max | 92.33% | 32.97% | 51.80% | 37.09% | 35.14% | 61.48% |
| Hunyuan-Vision | 53.78% | 33.33% | 49.73% | 33.33% | 35.51% | 58.45% |
| Hunyuan-Turbo-Vision | 84.56% | 37.32% | 50.00% | 33.33% | 41.67% | 59.51% |
| Step-1V-32K | 11.22% | 38.04% | 55.74% | 39.86% | 48.91% | 61.80% |
| Gemini-2.0-Pro-Exp | 96.00% | 40.58% | *Fail* | 38.04% | 56.88% | 60.63% |
| Qwen2-VL-2B | 4.11% | *Fail* | 46.72% | *Fail* | 18.84% | 62.00% |
| Qwen2-VL-7B | 22.56% | 31.16% | 52.46% | 31.88% | 26.45% | 61.54% |
| Qwen2-VL-72B | 67.11% | 24.64% | 50.55% | 28.62% | 36.59% | 61.84% |
| Deepseek-VL2 | 23.56% | 33.33% | 46.17% | 31.88% | *N/A* | 55.96% |
| Deepseek-VL | *Fail* | *Fail* | *Fail* | *Fail* | 28.99% | 58.99% |
| InternVL2.5-2B | 3.22% | 33.70% | *Fail* | 44.20% | 19.93% | 63.02% |
| InternVL2.5-8B | 40.78% | **42.39%** | 48.63% | **44.57%** | 28.99% | 62.31% |
| InternVL2.5-78B | 94.78% | 33.70% | 50.00% | 33.70% | **47.10%** | 62.01% |
| GLM-4V-9B | *N/A* | 35.51% | *N/A* | *Fail* | *N/A* | **63.53%** |
| LLaVA-OneVision-7B | 15.33% | *Fail* | *Fail* | *Fail* | 23.55% | 61.20% |
| MiniCPM-V-2.6-8B | 57.67% | *Fail* | *Fail* | 34.42% | 35.14% | 60.91% |
| Phi-3.5-vision | 48.11% | 31.52% | *Fail* | *Fail* | 23.91% | 57.00% |
| Ovis1.6-Gemma2-9B | 22.56% | 45.29% | *Fail* | 52.90% | *Fail* | 61.32% |
| Valley-Eagle-7B | 42.89% | 33.70% | *Fail* | 33.33% | 33.70% | 62.66% |
| MiniCPM-o-2.6-8B | 49.78% | 35.14% | 50.27% | 36.23% | 34.42% | 61.27% |
| Qwen2.5-VL-3B | 31.67% | 32.97% | *Fail* | *Fail* | 40.22% | 63.40% |
| Qwen2.5-VL-7B | 79.78% | *Fail* | *Fail* | *Fail* | 33.33% | 62.41% |
| Qwen2.5-VL-72B | **95.33%** | 32.97% | **55.19%** | 32.25% | 43.12% | 61.54% |
| Claude-3.7 | **95.56%** | 31.88% | 51.37% | 42.03% | 49.28% | *Fail* |
| QVQ-72B | 38.67% | 33.70% | 40.71% | 23.19% | 33.33% | 57.83% |

Table 11: Overall performance of LMMs on *Cultural Study* tasks. The best LMM of each set is **bold**, the second-best is underlined.

## J  Extended Icon Recognition Discussion

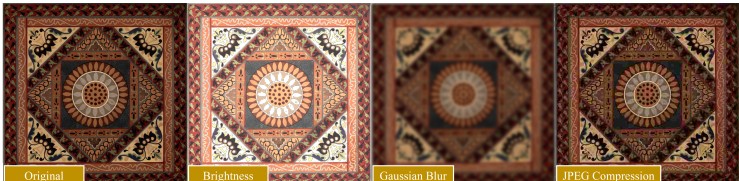

Figure 7: Illustration of image quality degradation.

| Model | Original | Brightness | Gaussian Blur | JPEG Compression |
|---|---|---|---|---|
| SenseNova | 93.88% | 93.00% | 84.54% | 92.93% |
| Gemini-2.0-Pro-exp | 97.00% | 95.00% | 93.00% | 98.00% |
| Qwen2.5-VL-72B | 93.00% | 27.00% | 31.00% | 28.00% |
| Qwen2.5-VL-7B | 76.00% | 25.00% | 24.00% | 25.00% |
| InternVL2.5-8B | 49.00% | 47.00% | 37.00% | 45.00% |

Table 12: Performance comparison under different conditions.

Due to variations in museum preservation status, digitization methods, and the historical aging of manuscripts, image quality variation are inevitable in real-world scenarios. To systematically assess how LMMs handle degraded visual inputs, we use Iconographic task as testbed for image quality degradation analysis. As shown in Figure 7, we introduce three conditional degradation types: Brightness Variation, Gaussian Blur, Image Compression Artifacts. We evaluate two closed-source LMMs (`SenseNova` and `Gemini-2.0-Pro-Exp`) and two open-source LMMs (`Qwen2.5-VL-72B` and `InternVL-8B`) with best performances on this task.

Results in Table 12 reveal that mainstream closed-source models exhibit more acceptable robustness than open-source LMMs. However, `Qwen2.5-VL-72B` shows a severe performance drop across all three degradation types. To further investigate, we tested `Qwen2.5-VL-7B`, which shares the same visual encoder but a different LLM backbone. The similar performance decline across noise conditions suggests that the performance drop likely originate from intrinsic limitations in the LMM's visual module during training, rather than from differences in language model backbone.

## K  Extended Chronological Attribution Discussion

Chronological Attribution is a specialized and complex task, even for human experts, often requiring extensive multi-expert verification and cross-referencing to determine the correct time period. Such cross-domain multimodal inference tasks are well-suited for LMMs to generate explanations or serve as reference tools. This task's inherent complexity motivate our multiple question formats design.

Experiments show that most current LMMs struggle with this task, with common failure cases including: predicting the same era for all images (typically defaulting to Tang Dynasty), irrelevant responses (particularly caused by CoT-based questions), incorrect reasoning paths or produce vague, uninformative justifications.

We observed notable failure patterns when switching from free-form Q&A to Binary comparison (determining whether one artifact predates another). Five small LMMs defaulted to predict a single era for all visual inputs. Theoretically, binary questions should be easier than direct period classification. As a result, these LMMs were proven engaging in random guessing.

To mitigate this, we incorporated explicit textual descriptions of stylistic features associated with different historical periods within CoT reasoning prompts, expecting LMMs to derive the correct answers through stepwise inference. However, for this holistic style perception task, CoT proved counterproductive. LMMs lacking pre-trained historical knowledge even generated identical CoT reasoning sequences for all images (e.g., `InternVL2.5`). Identical phenomenon observed in *Calligraphy Style Classification* validated our third key insights suggested in Section 1.

Our findings confirm that directly providing few-shot exemplars through visual retrieval augmentation yield the most significant and cost-efficient performance gains, making it the preferred approach for extrapolate LMMs' knowledge through analogy. This question format also allows us to identify substantial differences in models' cross-image perception abilities: LMMs that exhibit performance drops typically have weaker visual processing, modality bias or unsatisfactory cross-modal alignment.

## L    Extended Artwork Caption Discussion

We initially adopted BERTScore to remain consistent with prior LLM-for-culture studies [15, 20]. We also considered standard metrics for semantic similarity, including METEOR and ROUGE-L.

Our ground truth annotation process was specifically designed to capture historical and cultural nuance. Beyond common captioning elements (e.g., objects, scenes, and actions), our captions deliberately incorporate culture-specific features. To address the limitations of generic semantic similarity metrics in evaluating culturally grounded tasks, we incorporated four domain-informed evaluation dimensions in Table 13. **Religious Symbolism**: identifying and describing symbolic entities (e.g., Buddhas, Guardians, Apsaras). **Facial Expression**: distinguishing emotional or spiritual expression aligned with iconographic conventions (e.g., Buddhas are often compassionate, Guardians are more solemn, Flying Apsaras are relatively joyful). **Historical Period**: referencing the appropriate historical timeframe. **Event Specificity**: capturing culture-related events. These results reflect substantial variation in models' abilities to capture symbolic aspects of manuscript images. We hope these dimensions can serve as a starting point toward developing more structured and culture-aware evaluation metrics tailored for cultural related tasks.

| Model | METEOR | ROUGE-L | Symbol | Facial | Period | Event |
|---|---|---|---|---|---|---|
| GPT-4o | 0.1285 | 0.1391 | 0.6818 | 0.8696 | *Fail* | *Fail* |
| Step-1o-vision-32k | 0.1287 | 0.1109 | 0.9091 | 0.9565 | 0.4737 | 0.3333 |
| Gemini-2.0-Pro-Exp | 0.1110 | 0.0702 | 1.0000 | 0.9130 | 0.2105 | 0.5000 |
| JT-VL-Chat | 0.1282 | 0.1699 | 0.8182 | 0.8696 | *Fail* | *Fail* |
| Qwen2.5-VL-72B | 0.1303 | 0.1146 | 0.8636 | 0.9565 | *Fail* | 0.1667 |
| InternVL2.5-78B | 0.1302 | 0.1222 | 0.8636 | 0.9565 | 0.1053 | *Fail* |
| MiniCPM-o-2.6-8B | 0.1273 | 0.1304 | 0.8182 | 0.7391 | 0.0526 | *Fail* |

Table 13: Culture-aware evaluation results for Artwork Caption task. "*Fail*" indicates missing or incorrect information.

## M    Limitations

Due to the inherent scarcity of annotated data in manuscript research, we inevitably face challenges in dataset diversity, despite leveraging the most extensive and well-documented Dunhuang corpus. The primary limitations include:

**1) Limited cultural scope:** our current benchmark focuses on the Dunhuang corpus, our most accessible and well-curated source, supported by longstanding collaborations with domain experts. This focus allows us to construct high-quality tasks grounded in archaeological workflows and enable rigorous evaluation. We regard this as a first concrete step toward the broader goal of advancing *Ancient Manuscript Study* as a systematic and scalable research direction. While this single-source scope limits immediate cross-cultural generalization, many tasks in MS-Bench (e.g., *Fragment Restoration*, *Textual Recognition*, and *Cultural Study*) reflect shared challenges across manuscript traditions such as papyri, Dead Sea Scrolls, and medieval Bible copies. We thus position MS-Bench as a reusable and extensible framework, offering transferable task structures, prompting strategies, and evaluation protocols applicable to other cultural heritages in future extensions.

**2) Inherent data imbalance:** we acknowledge the existence of data imbalance among tasks and the long-tail phenomenon in specific sub-tasks (e.g., allograph normalization). This distribution reflects the real-world characteristics of manuscript collections rather than an artificial bias introduced during benchmark curation. Future iterations may explore balancing strategies while preserving authenticity and historical fidelity.

**3) Challenges in evaluating cultural understanding:** assessing LMMs' cultural reasoning ability remains challenging. Simply averaging scores across *Cultural Study* tasks fails to capture cultural biases in model predictions. We introduced expert evaluation for the *Artwork Caption* task and further expanded culture-aware evaluation metrics and structured criteria. Nonetheless, developing scalable and more systematic evaluation frameworks for cultural reasoning requires further exploration.

## N    Future Work

Due to data limitations, the current benchmark dataset is insufficient to directly fine-tune LMMs for historical manuscript study. In future work, 1) we aim to curate larger, expert-annotated corpora in collaboration with archaeologists and historians to enable the development of domain-specialized LMMs and tailored applications. 2) We may explore extending the current MS-Bench framework beyond the Dunhuang corpus. Many of our task designs reflect common challenges across manuscript traditions. Building on the modular structure and prompting strategies of MS-Bench, we hope to adapt and validate the benchmark in diverse cultural contexts under expert supervision. We aim to broaden MS-Bench to those underexplored sources (e.g., recently unearthed Xinjiang manuscripts, similar to Dunhuang, could directly benefit from the shared tasks defined in our benchmark). 3) Real-world user studies with archaeologists and manuscript scholars may help to assess the practical utility, usability, and interpretability of LMMs in field-specific workflows. We understanding this will bridge AI research with the human-in-the-loop needs of historical disciplines. Currently, we provide "model recommendations" and "recommended prompts" for archaeologists in our project page: https://github.com/ianeong/MS-Bench.

## O    Social Impact

Applying AI models in archaeology has gained increasing attention in recent years, with a growing number of interdisciplinary publications. Our collaboration with philologists revealed that while some archaeologists are highly interested in applying advanced AI techniques like LMMs, they often lack clear guidance for practical implementation (e.g., which model to use, how to write effective prompts). Our work aims to bridge this gap, providing a benchmark that highlights effective LMMs tailored for specific manuscript analysis tasks. We also hope to encourage AI researchers to explore more impactful and diverse applications of LMMs in archaeology and beyond. For example, incorporating richer cultural diversity and historical significative data into LMM pre-training and evaluation to facilitate new archaeological findings. Moving forward, we seek to further investigate human-AI collaboration frameworks that can effectively integrate domain expertise with AI capabilities in manuscript research.

On the negative side, we recognize that the fundamental hallucination issues inherent in current LMMs, which could prevent philologists from trusting model outputs. Furthermore, enhancing model output interpretability remains a crucial challenge, as some scholars may be reluctant to adopt AI-generated or AI-assisted findings without clear and comprehensive historical evidence. Thus, LMMs should serve as assistive tools requiring expert oversight and domain-specific knowledge to ensure reliable collaborative outcomes.

