# Appendix: Examples from Benchmark

**Example 1: Handwritten Character Recognition**

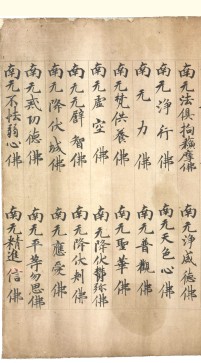

(a) Input Image

You are an expert in recognizing ancient texts. This is an image of a vertically oriented, handwritten ancient text in traditional Chinese characters. Please use your visual capabilities to perform OCR and extract the text, and only provide me with the OCR result.

**Answer**

南无法俱拘苏摩佛
南无净威德佛
南无净行佛
南无天色心佛
南无力佛
南无普观佛
南无梵供养佛
南无圣华佛
南无虚空佛
南无降伏郁弥佛
南无无臂智佛
南无降伏刺佛
南无降伏城佛
南无应爱佛
南无戒功德佛
南无平等勿思佛
南无不怯弱心佛
南无精进信佛

**Example 2: Allograph Normalization**

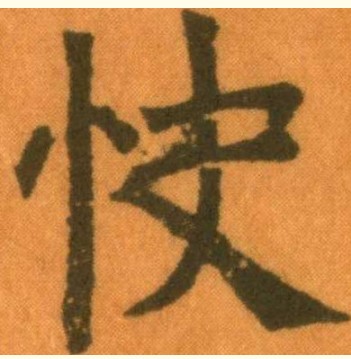

(a) Input Image

Please normalize the allograph of this character in this image.

快

**Example 3-1: Writing Symbol Detection**

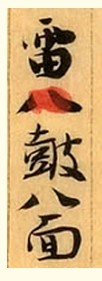 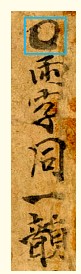 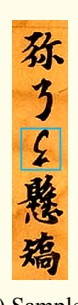 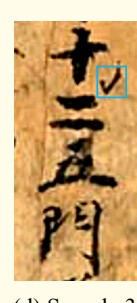 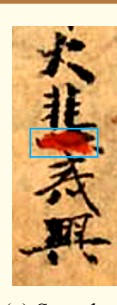 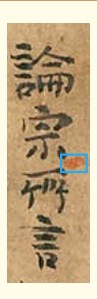

  (a) Input      (b) Sample 1     (c) Sample 2     (d) Sample 3    (e) Sample 4     (f) Sample 5

**Multiple-choice:**
Assume you are a specialist in bibliography, examining a calligraphy scroll for distinctive symbols. Please identify which specific symbol is most likely represented in the image below. Choose one from the following options: "Hierarchy Symbol", "Repetition Mark Symbol", "Tick Symbol", "Deletion Symbol", or "Pause Symbol". If none of these are applicable, please output "Not found."

**Chain-of-Thought (CoT):**
Assume you are a specialist in bibliography, examining a calligraphy scroll for distinctive symbols. Please identify which specific symbol is most likely represented in the image below. Choose one from the following options: "Hierarchy Symbol", "Repetition Mark Symbol", "Tick Symbol", "Deletion Symbol", or "Pause Symbol". If none of these are applicable, please output "Not found."
<Thinking Steps>
Step 1. Identify the part of the text in the image, and search for possible special symbols in the areas outside or on the text itself.
Step 2. Analyze the characteristics, functions, and possible positions of the symbols: hierarchical symbol, repetition mark symbol, tick symbol, deletion symbol, and pause symbol.
Step 3. Compare the image characteristics with the special symbol characteristics and make an overall judgment about which special symbol is most likely present.
Step 4. Choose the most likely symbol from "Hierarchy Symbol", "Repetition Mark Symbol", "Tick Symbol", "Deletion Symbol" and "Pause Symbol" and output the result. If no symbol is found, output "Not Found".

**Knowledge Retrieval:**
Assume you are a specialist in bibliography, examining a calligraphy scroll for distinctive symbols. Here are some examples, with the special symbols highlighted in blue boxes.
1. The image below contains the "Hierarchy Symbol": <sample1>
2. The image below contains the "Repetition Mark Symbol": <sample2>
3. The image below contains the "Tick Symbol": <sample3>
4. The image below contains the "Deletion Symbol": <sample4>
5. The image below contains the "Pause Symbol": <sample5>
Please identify which specific symbol is most likely represented in the image below. Choose one from the following options: "Hierarchy Symbol", "Repetition Mark Symbol", "Tick Symbol", "Deletion Symbol", or "Pause Symbol". If none of these are applicable, please output "Not found."

Deletion Symbol

**Example 3-2: Writing Symbol Detection (Binary)**

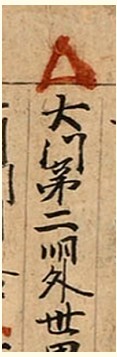

(a) Input Image

Please look for the following special symbol in the image: abbreviation symbol.
It is written in its original form, appears in black ink like the main text, functions as a re-placement for previously mentioned words, and is usually located below the annotations of repeated characters.
Please answer "Yes" or "No."

Answer

No

**Example 4: Calligraphy Style Classification**

### Prompt

(a) Input Image     (b) Sample 1     (c) Sample 2     (d) Sample 3     (e) Sample 4

**Multiple-choice:**

Please determine which Chinese calligraphy style is the closest to the one in the image. Please choose one from "Running Script", "Cursive Script", "Regular Script", and "Clerical Script" that is most similar to the image.

**Chain-of-Thought (CoT):**

Please determine which Chinese calligraphy style is the closest to the one in the image. Please choose one from "Running Script", "Cursive Script", "Regular Script", and "Clerical Script" that is most similar to the image.

<Thinking Steps>

Step 1. Analyze the image, observe the overall style, and initially classify it into broad categories, excluding some categories.

Step 2. Analyze the stroke characteristics of the four calligraphy styles and further exclude some.

Step 3. Evaluate the structural features of the four calligraphy styles.

Step 4. Make a comprehensive judgment, determine the calligraphy style that the image most closely resembles, and output the final answer.

**Knowledge Retrieval:**

This is an example of Running Script Calligraphy: <sample1>

This is an example of Cursive Script Calligraphy: <sample2>

This is an example of Regular Script Calligraphy: <sanple3>

This is an example of Clerical Script Calligraphy: <sample4>

Please determine which Chinese calligraphy style is the closest to the one in the image. Please choose one from "Running Script", "Cursive Script", "Regular Script", and "Clerical Script" that is most similar to the image.

### Answer

Cursive Script

**Example 5: Manuscript Damage Assessment**

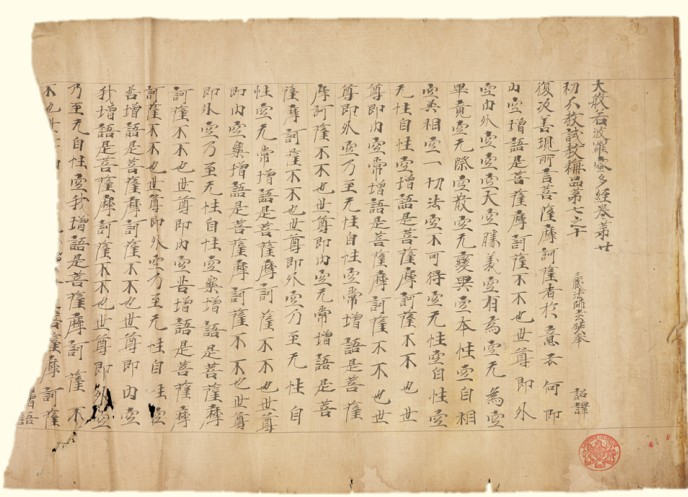

(a) Input Image

Please choose one of the four options below to assess the damage severity.
A. Left edge intact, right edge intact
B. Left edge damaged, right edge intact
C. Left edge intact, right edge damaged
D. Left edge damaged, right edge damaged

C. Left edge intact, right edge damaged

**Example 6: Fragmented manuscript Restoration**

**Example 6-1: Yes/no Judgement**

> **Prompt**
>
> 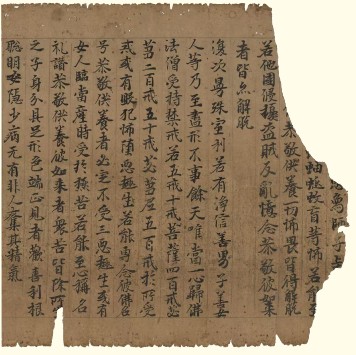 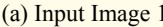 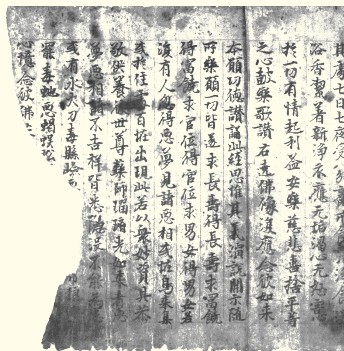
>
> (a) Input Image 1        (b) Input Image 2
>
> Assume you are an expert in fragment restoration, repairing these pieces like a jigsaw puzzle. I will provide you with two fragment images. Please help me determine whether these two pieces can be joined together.

> **Answer**
>
> Yes

**Example 6-2: Likelihood Estimation**

> **Prompt**
>
> 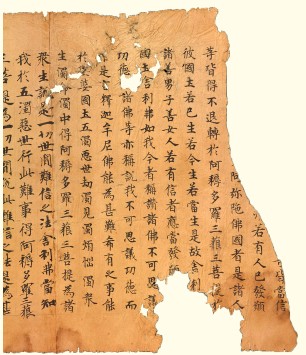 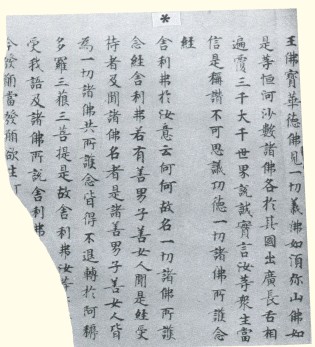
>
> (a) Input Image 1        (b) Input Image 2
>
> Assume you are an expert in fragment restoration, repairing these pieces like a puzzle. I will provide you with two fragment images. Please help me determine in which direction these two pieces should be joined based on the given images. Note: Do not write any code! Only use your visual ability. Please give a matching score from 0 to 100, where 100 means they can be completely joined together.

> **Answer**
>
> 100

**Example 6-3: Relative Positioning Hypotheses**

### Prompt

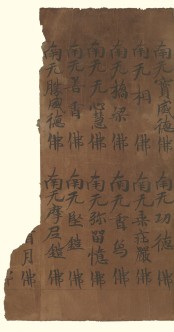

(a) Input Image 1

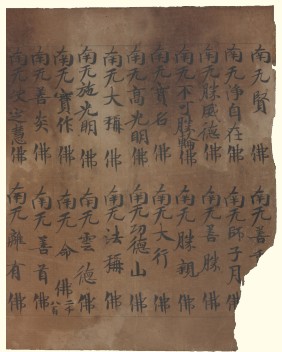

(b) Input Image 2

Assume you are an expert in fragment restoration, repairing these pieces like a jigsaw puzzle. I will provide you with two fragment images. Please help me determine in which direction these two pieces should be joined based on the given images.

### Answer

Fragment 1 is on the right, fragment 2 is on the left.

**Example 7: Icon Recognition**

### Prompt

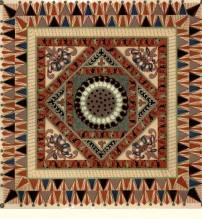 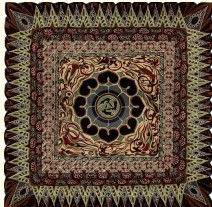 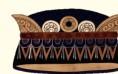 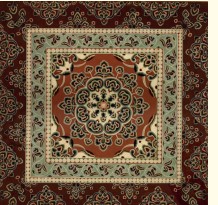

(a) Input Image 1   (b) Input Image 2   (c) Input Image 3   (d) Input Image 4

These are four images of Dunhuang decorative patterns: three of them are Zaojing (caisson ceiling) patterns, and one is a Huagai (canopy) pattern. Please distinguish and output the category of each image in sequence.

### Answer

The pattern of Zaojing; The pattern of Zaojing; The pattern of Huagai; The pattern of Zaojing.

**Example 8: Chronological Attribution**

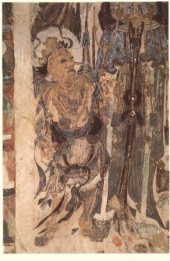 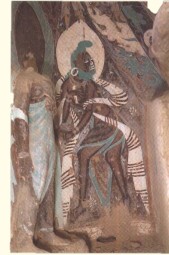 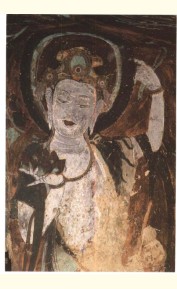 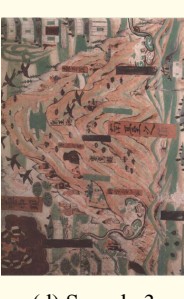 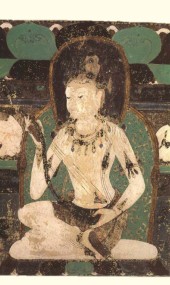

(a) Input Image     (b) Sample 1     (c) Sample 2     (d) Sample 3     (e) Sample 4

**Multiple-choice:**
This is a Dunhuang mural. Please determine the Chinese dynasty it most closely resembles. Choose the most appropriate period from the following options: "Tang Dynasty", "Wei-Jin and Northern and Southern Dynasties", "Five Dynasties", or "Yuan Dynasty".

**Chain-of-Thought (CoT):**
This is a Dunhuang mural. Please determine the Chinese dynasty it most closely resembles. Choose the most appropriate period from the following options: "Tang Dynasty", "Wei-Jin and Northern and Southern Dynasties", "Five Dynasties" and "Yuan Dynasty".
<Thinking Steps>
Step 1. Analyze the characteristics of Dunhuang murals from the Wei-Jin and Northern and Southern Dynasties, Five Dynasties, Tang Dynasty, and Yuan Dynasty in terms of figure modeling, clothing styles, color usage, and other aspects.
Step 2. Identify the figures in the mural, and analyze the characteristics of the figures' modeling, clothing styles, and color usage in the image.
Step 3. Compare the characteristics of the image with the Dunhuang mural characteristics described in Step 1 and Step 2, and make an overall judgment of which period the image's features are closest to.
Step 4. Choose the most similar period from "Tang Dynasty", "Wei-Jin and Northern and Southern Dynasties", "Five Dynasties" and "Yuan Dynasty" and provide the output.

**Knowledge Retrieval:**
Here are several Dunhuang murals from different periods.
Image 1 is from the period of "Wei-Jin and Northern and Southern Dynasties": <sample1>
Image 2 is from the period of "Tang Dynasty": <sample2>
Image 3 is from the period of "Five Dynasties": <sanple3>
Image 4 is from the period of "Yuan Dynasty": <sample4>
Based on the above information, please infer and determine the Chinese dynasty the following image belongs to. Choose the most appropriate period from the following options: "Tang Dynasty", "Wei-Jin and Northern and Southern Dynasties", "Five Dynasties" and "Yuan Dynasty".

Tang Dynasty.

**Example 8-2: Binary**

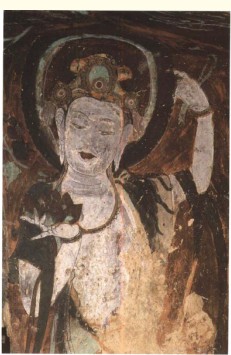
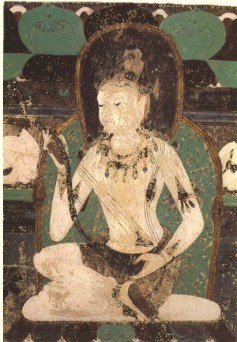

(a) Input Image 1
(It is from Tang Dynasty)

(b) Input Image 2
(It is from Yuan Dynasty)

Below are two Dunhuang murals from different periods. Please determine whether the date of Image 1 is earlier than that of Image 2.
Please answer "Yes" or "No".

**Answer**

Yes

**Example 9: Artwork Captioning**

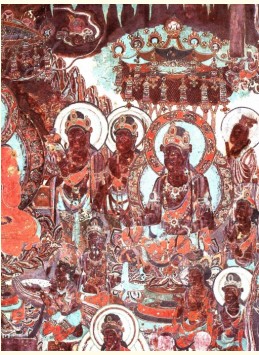

(a) Input Image

This is an mural of Dunhuang Grottoes. Please describe the content of the image with an appreciative tone.

This image features nine bodhisattvas, led by Manjushri, including Mahasthamaprapta, Akasagarbha, Bhaisajyaraja, Baoyue, Manjushri, Wuliangli, Badawopala, Baoji, and others. They are either seated or standing, arranged in a staggered manner with graceful postures. Their dark brown bodies are adorned with luxurious silk garments, jeweled necklaces, and intricate ornaments. The contrast of vibrant colors in red and green, especially with the bright halos around them, makes the image particularly striking and unique. Behind Manjushri is the inscription of the glorious and dignified king, with a crown for long travels, a tablet, and deep robes, presenting a distinct Chinese imperial attire.