# OpenReview forum: "MS-Bench: Evaluating LMMs in Ancient Manuscript Study through a Dunhuang Case Study"
_NeurIPS.cc/2025/Datasets_and_Benchmarks_Track — NeurIPS 2025 Datasets and Benchmarks Track poster_

### Official Review · Reviewer_Ynz3 · 2025-06-25

**Rating:** 4
**Confidence:** 5

**Summary:**

This paper introduces a novel multimodal benchmark for ancient manuscript study, built upon the Dunhuang dataset. The authors meticulously designed various test tasks to reflect genuine needs in linguistic studies. The dataset creation involved extensive research and careful design. Using this benchmark, the authors conducted evaluations on several large models, presenting their findings and a concise analysis. This research offers valuable guidance for the field of AI4Humanity.

**Additional Feedback:**

Nan

**Dataset Code Accessibility:**

Yes

**Ethical Considerations:**

No, there are no or only very minor ethics concerns

**Final Justification:**

Most of my concerns have been addressed, and I maintain a positive opinion.

**Limitations Weaknesses:**

1. The benchmark exclusively utilizes Dunhuang-related ancient manuscripts. This single-source focus may limit its representativeness for the broader domain of ancient texts.

2. The authors should further investigate the underlying reasons for the impact of CoT and V-RAG on model performance across these tasks, potentially through illustrative examples.

3. The authors could perform a more detailed analysis of the manuscript, such as the dynasty (different writing styles and fonts).

**Strengths Contributions:**

1. The process for organizing and annotating the Dunhuang data is of exceptionally high quality.

2. The benchmark features seven distinct major tasks, each comprising multiple sub-tasks. These tasks effectively represent key research directions in ancient manuscript studies.

3. The three research questions posed in the introduction are highly relevant and are substantiated with empirical evidence in the experimental section.

4. The supplementary materials provide comprehensive explanations for each task, enhancing the clarity and usability of the benchmark.

---

> ### Author Rebuttal · Authors · 2025-07-31
>
> We express our sincere gratitude to your comprehensive comments and thorough assessment of our paper. We appreciate the opportunity your questions provide to further clarify your concerned questions.
>
> > ***Q1: The benchmark exclusively utilizes Dunhuang-related ancient manuscripts. This single-source focus may limit its representativeness for the broader domain of ancient texts.***
>
> We appreciate your comment regarding the representativeness of our benchmark, which currently focuses exclusively on Dunhuang-related manuscripts. We agree that relying on a single source may raise concerns about generalizability across the broader landscape of ancient texts. That said, we respectfully note that the Dunhuang corpus itself spans a wide range of topics—from religious scripture and administrative documents to daily-life records and artistic imagery—and has long served as a global site of interdisciplinary scholarship in history, philology, art history, and archaeology. Many of the fundamental research challenges it poses in paleography, codicology, and iconography also appear in other manuscript traditions, such as fragment reassembly for the Dead Sea Scrolls and papyri, recognition of script variation in ancient Japanese manuscripts, and interpretation of visual symbolism in Bible copies. **In addition, we believe our benchmark design—hierarchical task structure, cognitively guided task decomposition, and multiple prompting strategies—may offer useful methodological insights for constructing similar datasets in other archaeological and cultural heritage domains**.
>
> Our decision to begin with Dunhuang was motivated by both academic and practical considerations: the availability of well-digitized, expert-annotated materials; and our team’s longstanding collaborations with domain scholars. We acknowledge that constructing benchmarks for ancient manuscript analysis presents several core challenges—**the scarcity of accessible high-quality data, the interdisciplinary gap between AI research and philological expertise, and the cultural and material heterogeneity across manuscript traditions**. MS-Bench represents an initial effort to address these challenges by establishing a reproducible and extensible framework. We are committed to actively maintaining and expanding this benchmark, and we welcome future collaborations to extend it to additional manuscript traditions and cultural settings.
>
> > ***Q2: The authors should further investigate the underlying reasons for the impact of CoT and V-RAG on model performance across these tasks, potentially through illustrative examples.***
>
> We appreciate your suggestion to further investigate the underlying reasons, we agree that illustrative examples would offer more valuable insights.
>
> In general, **CoT is more effective when task benefits from explicit step-wise reasoning and disambiguation** (e.g., Writing Symbol Detection). In contrast, for perception-heavy tasks requiring holistic judgement or visual intuition (e.g., Calligraphy Style Classification and Chronological Attribution), CoT may induce over-analysis or model hesitation.
> For example, in Calligraphy Style Classification, we observed that CoT prompts often led models such as Gemini or InternVL to generic or default answers such as “Regular Scripts” or even “I am not sure”, due to a misalignment between expert-designed complex reasoning logic and model’s internal visual judgment. It is inherently difficult to “teach” models to differentiate subtle holistic patterns through textual descriptions alone (e.g., between Clerical and Running Script), especially without pretraining on such domain-specific stylistic variations. On the other hand, in Writing Symbol Detection, CoT helps models correctly disambiguate visual marks. For example, CoT guides model to to focus on regions outside the main text area. This helps distinguish between small marks like “Tick Symbol” and a larger similar shaped character or strokes.
>
> Compared to CoT, **V-RAG offers more stable an interpretable improvements by grounding model predictions with visual exemplars**. For example, in Writing Symbol Detection, model directly “compares” given manuscript with reference samples to locate matching patterns, benefiting from its inherent visual grounding capability without exterior domain knowledge required. In Chronological Attribution, for example, both “Tang Dynasty” and “Wei-Jin Dynasty” figures may exhibit richer color palettes, but “Tang” figures are characteristically rounder in facial and body structure. V-RAG allows models to align unknown inputs with these reference traits, improving relative dating accuracy without relying solely on unstable textual reasoning.
>
> In addition to the extended discussions provided in Appendix E~I, we will include more illustrative, case-based analyses in the final version to clearly highlight these patterns. We are also actively collaborating with archaeologists to further investigate model failure cases and find the “best practice” for non-AI specialists working in historical document analysis.
>
> > ***Q3: The authors could perform a more detailed analysis of the manuscript, such as the dynasty (different writing styles and fonts).***
>
> Thank you for your helpful suggestion! We have now added more detailed analyses of manuscript dynasties, writing styles (fonts), image sources, manuscript status, and the distribution of writing symbols (please refer to the updated table). We respectfully note that some statistics are not available for every image in our dataset. For example, the dating of certain manuscripts used in the Material Study task remains under debate among archaeologists, so for consistency and accuracy, we only include manuscripts used in Chronological Attribution tasks when reporting dynasty-level statistics. We also acknowledge an imbalance in calligraphic style distribution—this reflects the real-world composition of the Dunhuang manuscripts, in which over 90% of manuscripts are Buddhist scriptures written in Regular Script. These additional statistics will be included in the revised PDF to enhance transparency and reproducibility.
>
>   **Dimension &nbsp; &nbsp; &nbsp; &nbsp; &nbsp; &nbsp; &nbsp; Statistics**
>   ***
>
>   Image Source &nbsp; &nbsp; &nbsp; &nbsp; &nbsp;54.64% Specialist Collection &nbsp; 24.99% Publications & Research Paper &nbsp; 12.81% Museum &nbsp; &nbsp; &nbsp; &nbsp; &nbsp; &nbsp; &nbsp; 7.56% Others
>
>   Writing Symbol &nbsp; &nbsp; &nbsp; 48.46% Deletion &nbsp; &nbsp; &nbsp; &nbsp; &nbsp; &nbsp; &nbsp; &nbsp; &nbsp; &nbsp; &nbsp; &nbsp;19.90% Repetition &nbsp; &nbsp; &nbsp; &nbsp; &nbsp; &nbsp; &nbsp; &nbsp; &nbsp; &nbsp; &nbsp; &nbsp; &nbsp; &nbsp; &nbsp; &nbsp; &nbsp; &nbsp; &nbsp; 18.28% Hierachy &nbsp; &nbsp; &nbsp; &nbsp; &nbsp; &nbsp; &nbsp; 6.71% Pause &nbsp; &nbsp; &nbsp; &nbsp; &nbsp; &nbsp; &nbsp; &nbsp; &nbsp; &nbsp; &nbsp; 6.63% Tick
>
>   Calligarphy Style &nbsp; &nbsp; 91.38% Regular Script &nbsp; &nbsp; &nbsp; &nbsp; &nbsp; &nbsp; &nbsp;2.87% Running Script &nbsp; &nbsp; &nbsp; &nbsp; &nbsp; &nbsp; &nbsp; &nbsp; &nbsp; &nbsp; &nbsp; &nbsp; &nbsp; &nbsp; &nbsp; &nbsp; 2.87% Clerical Script &nbsp; &nbsp; &nbsp; &nbsp; 2.87% Cursive Script
>
>   Dynasty &nbsp; &nbsp; &nbsp; &nbsp; &nbsp; &nbsp; &nbsp; &nbsp; &nbsp; &nbsp; 33.33% Tang Dynasty &nbsp; &nbsp; &nbsp; &nbsp; &nbsp; &nbsp; &nbsp;33.33% Wei-Jin Dynasty &nbsp; &nbsp; &nbsp; &nbsp; &nbsp; &nbsp; &nbsp; &nbsp; &nbsp; &nbsp; &nbsp; &nbsp; &nbsp; &nbsp; &nbsp;16.67% Five Dynasties &nbsp; &nbsp; 16.67% Yuan Dynasty
>
>   Manuscript Status &nbsp; 64.91% Damaged &nbsp; &nbsp; &nbsp; &nbsp; &nbsp; &nbsp; &nbsp; &nbsp; &nbsp; &nbsp; 35.09% Intact
>
> We hope that the above response will fully address your concerns. Thank you again for your thorough and helpful review! We eagerly anticipate your further suggestions and thoughts.

---

### Official Review · Reviewer_dWUb · 2025-06-29

**Rating:** 5
**Confidence:** 4

**Summary:**

This paper provides 5k images on archeology along with 10k questions to answer on them. It evaluates 32 LLMs perform against human raters across four prompting strategies.
It aims to help archaeologist better understand in domain pictures by providing a benchmark for research to build high performing ML methods.
It presents insights into the performance of LLMs on this benchmark and how prompting techniques (CoT, retrieval augmented) impact the results.

**Additional Feedback:**

Thank you for the contribution of this dataset and benchmark, it will help improve the performance of LLMs on ancient manuscrit.

I have some questions / follow ups that might be interesting to mention in the paper:
1. Do you think the methods presented in the paper to build such a dataset would extend to other manuscrits (say Arabic or Sanskrit) ?
2. Can you say if you observed that some prompting strategy worked better on some task types? Do you have any recommendation for other benchmark builders?
3. How do you think this kind of LLM use could help archeologists directly ?

**Dataset Code Accessibility:**

Partly

**Dataset Code Comments:**

The dataset is available at https://doi.org/10.7910/DVN/MKRTMN
The benchmark code is available at https://anonymous.4open.science/r/NeruIPS-DnB-Submission-1201-MS-Bench

The benchmark code is extremely partial. It contains only a simple call to an LLM API but no details on dataset loading, metrics implementation, ...

The dataset website is slow and the files are presented without explanation on what each file is or code to load it.

In the current state it would be difficult for interested readers to re-use the dataset or benchmark.

**Ethical Comments:**

No ethical concerns

**Ethical Considerations:**

No, there are no or only very minor ethics concerns

**Final Justification:**

I maintain a score of 5 (Accept). This is a strong, well-executed paper with clear value for both the AI and digital humanities communities.
Key Strengths:
- First large-scale benchmark on ancient manuscripts, with diverse tasks and high-quality data.
- Comprehensive evaluation (32 models, 4 prompting strategies, human baselines).
- Rebuttal meaningfully addresses concerns on generalizability, model anomalies, and usability.
- Ablations and prompting analysis add practical insights for future work.

Minor Remaining Limitations:
- No formal user study, but active expert collaborations and prompt guidance mitigate this.


Overall, this paper represents a valuable and timely contribution. I recommend acceptance.

**Limitations Weaknesses:**

- The benchmark provides 5k images and 10k questions; however it is centered on Dunhuang manuscrits which limits its potential applicability to other cultures and regions.
- The anomaly noted in section 3.1 of a smaller model overperforming a much larger one is noted but could be investigated further to better understand how this could be fixed.
- No user studies to understand how this could be directly used by archeologists is presented which limits its potential for helping archeologists.
- Ablation studies on the prompt style and task types may help understand the relative importance of these methods and improve how to build such benchmarks.

More generally, although such domain specific benchmarks are helpful to improve performance of LLMs on specific domains, this paper indeed limit itself to this specific domain and the applicability of the methods to a broader set of domains or tasks is not discussed.

**Strengths Contributions:**

The paper has several strength:

- It is one of the first papers to do an in depth analysis of ancient manuscrit tasks using LLMs and it goes beyond previous works (such as OBI-Bench and TimeTravel) by addressing more tasks (restoration)
- The quality of the labeling is high as it was co-built and verified with domain experts.
- The analysis is comprehensive with 32 LLMs evaluated and 4 different prompting strategies to probe the different kind of information that - are important for archeology.
- The task coverage is high; from ability to do OCR to cultural reasoning, with human baselines
- The paper is well organized with clear (although dense) figures.

---

> ### Author Rebuttal · Authors · 2025-07-31
>
> We are deeply grateful for your positive and encouraging feedback. We particularly appreciate your recognition of the novelty, the task coverage and data quality, evaluation, paper presentation. Your suggestions are highly valuable and will help us further advance our work to higher standards. Hereafter, we provide detailed responses to your concerns and questions.
>
> > ***Q1: It is centered on Dunhuang manuscripts which limits its potential applicability to other cultures and regions.***
>
> We appreciate your thoughtful comments. We agree that our current focus on Dunhuang may limit immediate applicability to other cultural contexts. Our goal is to contribute a **concrete and extensible step** in this emerging area. We began with the Dunhuang because it offers well-curated materials, established collaborations with domain experts, and a strong scholarly foundation—enabling us to construct high-quality, task-aligned benchmarks grounded in real philological workflows.
>
> We are also encouraged by the growing community interest, with recent works exploring oracle bones, historical artifacts, epigraphic inscriptions, etc. While our benchmark centers on the Dunhuang (which itself spans long historical period and diverse topics, and is globally recognized as an interdisciplinary research field), we hope it can support a broader movement to AI for archaeology by **offering transferable insights**—such as hierarchical task structures, prompting strategies, and evaluation protocols.
>
> Ultimately, our goal is not merely to evaluate current models, but to provide a **reusable and extensible framework** for building archaeological benchmarks. We view this as a long-term effort and are committed to actively maintaining and expanding MS-Bench as new cases and manuscript sources emerge. We hope this work contributes meaningfully to improving LMM capabilities while supporting real-world applications.
>
> > ***Q2: The anomaly noted in section 3.1 of a smaller model overperforming a much larger one is noted but could be investigated further to better understand how this could be fixed.***
>
> We appreciate your observation regarding this anomaly, which we agree is worth further discussion. Based on our analysis, this performance reversal stems from allograph type-specific sensitivities:
>
> 1. In script variation type, which involves subtle stroke-level differences, smaller models sometimes perform better–likely due to their reduced visual sensitivity, which makes them less susceptible to insignificant changes. In contrast, the larger models perform better on semantic component variation cases, where deeper reasoning about component-meaning alignment is needed. We propose that **collaborative inference** between large and small models may combine their complementary strengths to improve **overall accuracy**.
>
> 2. For the more challenging **radical repositioning** cases—where both models struggle due to substantial allograph deviation—we believe that **domain-specific continue pretraining** is needed. Given training **efficiency** constraints, smaller models may be more practical for targeted fine-tuning.
>
> We therefore view this as a **design opportunity**, where future systems can combine the **efficiency of smaller models** with the **stronger overall performance of larger models** to achieve more robust and adaptable solutions. Future research may include **model collaboration strategies, selective decoding**, or **fine-tuning smaller models under expert supervision**.
>
> > ***Q3: No user studies to understand how this could be directly used by archeologists is presented which limits its potential for helping archeologists.***
>
> We appreciate your comment regarding the absence of formal user studies. Based on your suggestion, we agree that incorporating large-scale user-centered evaluation would further enhance the practical value of our benchmark. We are currently engaged in on-going collaborations with our team’s archaeologists to analyze LMM outputs, particularly failure cases in OCR and allograph. They have proven valuable in domain experts’ view, who are often more interested in understanding where and why models fail.
>
> We also agree that providing actionable guidances would directly suppport archaeologists. In the revised Appendix, we would include the **“model recommendations”** and update **“recommended prompts”** for specific tasks. For example, structured prompts like CoT and visual-augmented formats tend to outperform generic instructions, which are still commonly used by archaeologists. We hope these additions will help bridge the gap between model capabilities and real-world direct usage.
>
> > ***Q4: Ablation studies on the prompt style and task types may help understand the relative importance of these methods and improve how to build such benchmarks.***
>
> > ***Additional 2: Can you say if you observed that some prompting strategy worked better on some task types? Do you have any recommendation for other benchmark builders?***
>
> We appreciate your interest in the relationship between prompting strategies and task types.
>
> For task types, CoT is effective in tasks that benefit multi-step reasoning and explicit disambiguation (e.g. Symbol Detection). Based on our empirical observations and iterative design, we offer the following insights:
>
> 1.	Role-playing is a simple yet effective strategy that improves task alignment;
> 2.	Providing **explicit reasoning steps or focal aspects** often works better than relying on the model’s own decomposition;
> 3.	**Clarifying task-specific terminology** helps reduce ambiguity.
>
> For **perception-heavy or holistic classification tasks** (e.g., Calligraphy and Chronological Attribution), CoT can sometimes lead to over-reasoning or model hesitation, resulting in vague or default responses. In such cases, visual references augmentation provides **more stable gains by anchoring responses in concrete visual exemplars** (e.g., CoT won’t help decipher unknow characters, but providing visual exemplar would help distinguish structural differences).
>
> For ablation across prompt styles, we tested different prompting strategies (simple role-play and few-shot reference) on Calligraphy Style Classification.
>
>   **Model &nbsp; &nbsp; &nbsp; &nbsp; Role-Play &nbsp; Few-shot**
>   ***
>
>   GPT-4o &nbsp; &nbsp; &nbsp; &nbsp; &nbsp; &nbsp; &nbsp;0.6767 &nbsp; 0.7467
>
>   Step-1o &nbsp; &nbsp; &nbsp; &nbsp; &nbsp; &nbsp; 0.7067 &nbsp; 0.7833
>
>   Qwen-max &nbsp; &nbsp; &nbsp; 0.7700 &nbsp; 0.7533
>
>   Qwen2.5-72B &nbsp; 0.7867 &nbsp; 0.8033
>
> We believe prompting is a crucial yet underexplored dimension of archaeological benchmark, given that some tasks can be prompt-sensitive. For future builders, we offer two recommendations:
>
> 1. Incorporate multiple strategies, and evaluate not only absolute performance but also interactions between prompts, task type, and model characteristics (e.g., hallucination resistance, visual grounding);
>
> 2. For culturally grounded tasks, consider co-designing prompts with domain experts and iteratively refining them.
>
> We will include these examples, ablations and case studies in the revised PDF.
>
> > ***Dataset Code Comments***
>
> We sincerely thank you for the comments regarding dataset and code usability. We have already made substantial improvements to address your concerns:
>
> 1. Full scripts including raw image preprocessing, dataset loading, inferencing, post-processing, metric implementation, etc.
>
> 2. Dataset is slow: We have already prepared mirrored uploads of  dataset to Google Drive, re-organized by zip archives.
>
> 3. Explanation and reusability: we have updated the Readme with implementation details.
>
> We hope these improvements will make it easier for both AI researchers and domain scholars to reuse and extend the benchmark. We remain committed to maintaining the dataset and codebase for ensuring broader impact and long-term usability.
>
> > ***Additional 1: Do you think the methods presented in the paper to build such a dataset would extend to other manuscripts (say Arabic or Sanskrit) ?***
>
> Thank you for this important question. While we recognize that different manuscript traditions pose distinct challenges, we believe the **design methodology presented in our paper is potentially transferable** to other traditions (Arabic, Latin, etc.). For Sanskrit, the situation may be more complex, given the limited OCR capabilities and LMM support for its script variants. Besides the scholarly-driven and hierarchical task design, many of the core tasks we defined (OCR, fragment restoration, style classification, etc.) are relevant across cultures, and the cognitive workflows of philologists share similarities across global traditions.
>
> > ***Additional 3: How do you think this kind of LLM use could help archeologists directly?***
>
> Thank you for this thoughtful question. We summarize below how our work is already supporting experts and how we plan to extend its impact.
> 1. Current Practical Value
> - MS-Bench provides clear, task-level comparisons of model performance and prompting strategies. Some experts in our team have begun constructing their own “prompt bases” based on our guidance and empirical findings.
> - Experts identify model limitations and failure cases that align with their research concerns (e.g., why models struggle with recognizing structurally transformed allographs).
>
> 2. Ongoing and Future directions
> -  We are currently exploring an LMM-centered toolkit involving different models for complex workflows. For example, restoration is a challenge that may benefit from model collaboration: shape-based matching models, LMM-based OCR, LLM-based semantic reasoning, etc.
> - We also aim to broaden MS-Bench to other underexplored sources (e.g., recently unearthed Xinjiang manuscripts—similar to Dunhuang—could directly benefit from the shared tasks defined in our benchmark).
>
> We look forward to your feedback and will be available for any of your inquiries at your earliest convenience.

---

> ### Comment · Reviewer_dWUb · 2025-08-08
>
> Thank you for the thoughtful and detailed rebuttal. Your clarifications addressed key concerns:
> - The focus on Dunhuang is justified by expert access and data quality, and your plan to generalize the methodology to other traditions is convincing
> - Your explanation around allograph-specific sensitivities and the potential for model collaboration was insightful and adds depth to your evaluation.
> - The ablation results and task-specific insights (e.g., CoT for reasoning, visual examples for classification) seem particularly useful and will help future benchmark builders.
> - Your improvements to dataset documentation and code structure will make the benchmark more accessible to both AI researchers and archaeologists.
> - While a formal user study is missing, your ongoing collaborations and concrete additions (like prompt recommendations) go a long way toward making the benchmark actionable.

---

> > ### Author Response · Authors · 2025-08-09
> > **Response to Reviewer dWUb**
> >
> > We are truly grateful for the your positive evaluation of our work. Your assessment is deeply motivating, and we sincerely appreciate the time and expertise you devoted to the review process.

---

### Official Review · Reviewer_3215 · 2025-07-01

**Rating:** 4
**Confidence:** 3

**Summary:**

This paper presents MS-Bench, the first comprehensive benchmark for evaluating large language models (LMMs) in ancient manuscript study. Developed in collaboration with archaeologists, MS-Bench consists of 5,076 high-resolution images from the 4th to 14th century and 9,982 expert-curated questions across nine sub-tasks. The benchmark evaluates 32 LMMs using four prompting strategies, revealing scale-driven performance improvements, the impact of prompting methods, and task-specific capabilities. While LMMs show promise in automating repetitive tasks, they still lack domain expertise, highlighting the need for human-AI collaboration.

**Dataset Code Accessibility:**

Yes

**Ethical Considerations:**

No, there are no or only very minor ethics concerns

**Final Justification:**

My concerns have been largely addressed by the rebuttal, and I would like to raise my score to borderline acceptance.

**Limitations Weaknesses:**

- MS-Bench's primary weakness is its exclusive focus on Chinese manuscripts, specifically from the Dunhuang collection. This narrow scope limits the generalizability of the findings to other ancient languages, scripts, and civilizations. Though the authors explicitly acknowledge this limitation in Appendix K, stating, "Our benchmark dataset is limited to Chinese manuscripts, lacking broader representation of other civilizations and script varieties". While the depth of analysis on Chinese manuscripts is a strength, the title's claim of a benchmark for "Ancient ManuScript Study" in a general sense is not fully supported. Also, the challenges in paleography, codicology, and iconography vary significantly across cultures.
- The evaluation of high-level cultural tasks, particularly "Artwork Caption," relies on metrics that may not fully capture the required historical and semantic nuance. While they commendably incorporate manual expert scoring to mitigate this, this process is subjective and difficult to scale. A more structured evaluation method is needed.

**Strengths Contributions:**

- The paper introduces MS-Bench, the first comprehensive and large-scale benchmark designed for ancient manuscript analysis. Its core strength lies in its scholarly-grounded design, which integrates 5,076 high-resolution images and 9,982 expert-curated questions into a hierarchical framework of nine sub-tasks that align with real-world archaeological workflows and cognitive reasoning levels.
- It provides a rigorous and systematic evaluation of 32 LMMs, employing four distinct prompting strategies to assess their effectiveness, robustness, and cultural grounding. This yields critical insights, such as the scale-driven reliability of larger models , the "two-sided effect" of Chain-of-Thought reasoning , and the consistent performance gains from retrieval augmentation.
- It is exceptionally well-presented. It is clearly written, logically organized, and easy to understand. The use of informative figures and tables effectively illustrates the benchmark's structure, the experimental pipeline, and the detailed performance results

---

> ### Author Rebuttal · Authors · 2025-07-31
>
> We sincerely thank the reviewer for the thoughtful and encouraging feedback on our benchmark design, evaluation strategy, and presentation. We appreciate the opportunity to further clarify specific concerns to ensure the methodology and contributions are communicated as clearly and completely as possible.
>
> > ***Q1: MS-Bench's primary weakness is its exclusive focus on Chinese manuscripts, specifically from the Dunhuang collection. This narrow scope limits the generalizability of the findings to other ancient languages, scripts, and civilizations. Though the authors explicitly acknowledge this limitation in Appendix K, stating, "Our benchmark dataset is limited to Chinese manuscripts, lacking broader representation of other civilizations and script varieties". While the depth of analysis on Chinese manuscripts is a strength, the title's claim of a benchmark for "Ancient ManuScript Study" in a general sense is not fully supported. Also, the challenges in paleography, codicology, and iconography vary significantly across cultures.***
>
> We appreciate the reviewer’s concern regarding our benchmark’s current focus on Dunhuang Manuscripts, and we would like to clarify the rationale behind this design choice and explain why we believe it represents a meaningful and extensible step toward broader goals in ancient manuscript analysis.
>
> 1. **Clarifying Scope and Positioning**: We fully acknowledge the reviewer’s concern regarding the limited cultural scope of our benchmark, and we agree that the challenges in paleography, codicology, and iconography vary across civilizations. Our intent is to take a **first concrete step** by focusing on the Dunhuang corpus—our most accessible and well-curated source, supported by existing collaborations with domain experts—to construct high-quality tasks grounded in archaeological workflows and enable rigorous evaluation. We hope this effort can serve as a foundation for our broader long-term vision of advancing Ancient Manuscript Study as a systematic and scalable research direction. That said, we acknowledge that our current scope may not fully reflect the diversity implied by the term, and we are open to revising the title to more accurately represent the focus (e.g., “…with Dunhuang as a Case Study”) if the reviewers and area chairs find it appropriate.
>
> 2. **Toward Generalization Across Cultures**: Our current benchmark is grounded in Dunhuang manuscripts, and we believe it offers **potentially transferable design insights** for constructing similar benchmarks in other manuscript traditions. Many tasks included in MS-Bench are not unique to Dunhuang, they reflect shared research needs across diverse manuscript traditions. For example, “Fragment Restoration” in codicology is also applicable to other broken manuscripts such as the Dead Sea Scrolls and ancient papyri. “Cultural Study” tasks involving visual iconography and artwork understanding also align with illustrated traditions such as medieval Bible copies. In paleography, “Textual Recognition” tasks could be extended to Japanese manuscripts, which were primarily written in Classical Chinese with minimal local annotation. Newly unearthed Xinjiang manuscripts—comparable in time period, material, and calligraphy style to Dunhuang—could adopt our current evaluation protocols with minimal adaptation. We agree that **cross-cultural generalization must be tested empirically under domain expert guidance**, and we view this as an important direction for future extension and validation.
>
> 3. **Cultural Breadth and Global Relevance of Dunhuang Manuscripts**: We respectfully note that the Dunhuang manuscripts holds significant cultural value and historical importance. It also provides **substantial internal diversity**, with a long historical span (from the 4th to 14th century) and diverse content (from religious texts, to administrative records and everyday documents). Its geographic position at the crossroads of East, Central, and South Asia fostered rich cultural exchange. Its close connection with cave murals, archaeological sites, and visual iconography further supports interdisciplinary research in history, art, and material culture. Today, Dunhuang materials are preserved in major institutions across the UK, France, China, Russia, and Japan, and the field of Dunhuang studies is widely recognized as an important area of international scholarship. Given the breadth, we believe the benchmark offers **meaningful relevance beyond its immediate cultural origin**, even being grounded in a specific manuscript tradition.
>
> 4.	**Community Contribution and Future Expansion**: High-quality, expert-curated datasets remain scarce yet critical for AI research in cultural heritage, where data collection and annotation are inherently complex. We are encouraged by the recent emergence of works focusing on specific materials or traditions (e.g., Epigraphy Inscriptions for textual restoration, Oracle Bone collections for recognition and restoration, fragmented artifacts for reassembly). Our goal is to position MS-Bench as a **reusable and extensible template**, with a modular task structure and prompting strategies that can **potentially be adapted** to other manuscripts. In addition to open-sourcing our dataset and code, we are committed to **actively maintaining and expanding MS-Bench** as new case studies and sources emerge—fostering future human–AI collaboration and advancing research across both AI and archaeology communities.
>
>
> > ***Q2: The evaluation of high-level cultural tasks, particularly "Artwork Caption," relies on metrics that may not fully capture the required historical and semantic nuance. While they commendably incorporate manual expert scoring to mitigate this, this process is subjective and difficult to scale. A more structured evaluation method is needed.***
>
> We appreciate your insightful feedback on evaluating high-level cultural tasks such as “Artwork Caption”. In designing our evaluation protocol, we initially adopted BERTScore to remain consistent with prior LLM-for-culture studies[1-2]. Similar to [2], we also considered standard metrics for semantic similarity, including METEOR and ROUGE-L.
>
> Our ground truth annotation process was specifically designed to capture **historical and cultural nuance**. Beyond common captioning elements (e.g., objects, scenes, and actions), our captions deliberately incorporate **culture-specific features**, such as religious symbolism (e.g., Buddhas), facial expressions (which carry significant meaning in religious contexts, e.g., Buddhas are often compassionate, Guardians are more solemn, Flying Apsaras are relatively joyful), and references to historical or ritual events. These elements are closely tied to iconography and help reflect the semantic depth needed for historical and cultural interpretation.
>
> We fully agree that more **structured and culture-aware evaluation metrics** tailored for cultural related tasks would further improve assessment quality. In addition to two standardized similarity-based metrics as [2], we have incorporated several **Culture Elements** dimensions, such as the correct identification of religious symbolism, facial expressions, historical periods, and event specificity. These domain-informed criteria will guide future metric design and annotation protocols, and we will elaborate in the revised PDF. We hope this can serve as a starting point for developing richer, more structured evaluation for culturally grounded tasks.
>
>   **Model &nbsp; &nbsp; &nbsp; &nbsp; &nbsp; &nbsp; &nbsp; &nbsp; &nbsp; &nbsp; &nbsp; &nbsp; &nbsp; METEOR &nbsp; &nbsp; ROUGE-L &nbsp; &nbsp; Symbol &nbsp; &nbsp; Facial &nbsp; &nbsp; Periods &nbsp; &nbsp; Event**
>   ***
>
>   GPT-4o &nbsp; &nbsp; &nbsp; &nbsp; &nbsp; &nbsp; &nbsp; &nbsp; &nbsp; &nbsp; &nbsp; &nbsp; &nbsp; 0.1285 &nbsp; &nbsp; &nbsp; &nbsp; 0.1391 &nbsp; &nbsp; &nbsp; &nbsp; 0.6818 &nbsp; &nbsp; &nbsp;0.8696 &nbsp; &nbsp; &nbsp; &nbsp;Fail &nbsp; &nbsp; &nbsp; &nbsp; &nbsp; &nbsp;Fail
>
>   Step-1o-vision-32k &nbsp; &nbsp; &nbsp; 0.1287 &nbsp; &nbsp; &nbsp; &nbsp; 0.1109 &nbsp; &nbsp; &nbsp; &nbsp; 0.9091  &nbsp; &nbsp; &nbsp;0.9565 &nbsp; &nbsp; 0.4737 &nbsp; &nbsp; &nbsp;0.3333
>
>   Gemini-2.0-Pro-Exp &nbsp; &nbsp; 0.1110 &nbsp; &nbsp; &nbsp; &nbsp; 0.0702 &nbsp; &nbsp; &nbsp; &nbsp; 1.000 &nbsp; &nbsp; &nbsp; &nbsp; 0.9130 &nbsp; &nbsp; 0.2105 &nbsp; &nbsp; &nbsp;0.5000
>
>   JT-VL-Chat &nbsp; &nbsp; &nbsp; &nbsp; &nbsp; &nbsp; &nbsp; &nbsp; &nbsp; &nbsp; &nbsp;0.1282 &nbsp; &nbsp; &nbsp; &nbsp; 0.1699 &nbsp; &nbsp; &nbsp; &nbsp; 0.8182 &nbsp; &nbsp; &nbsp;0.8696 &nbsp; &nbsp; &nbsp; &nbsp;Fail &nbsp; &nbsp; &nbsp; &nbsp; &nbsp; &nbsp;Fail
>
>   Qwen2.5-VL-72B &nbsp; &nbsp; &nbsp; &nbsp; &nbsp;0.1303 &nbsp; &nbsp; &nbsp; &nbsp; 0.1146 &nbsp; &nbsp; &nbsp; &nbsp; 0.8636 &nbsp; &nbsp; &nbsp;0.9565 &nbsp; &nbsp; &nbsp; &nbsp; Fail &nbsp; &nbsp; &nbsp; &nbsp; 0.1667
>
>   InternVL2.5-78B &nbsp; &nbsp; &nbsp; &nbsp; &nbsp; 0.1302 &nbsp; &nbsp; &nbsp; &nbsp; 0.1222 &nbsp; &nbsp; &nbsp; &nbsp; 0.8636 &nbsp; &nbsp; &nbsp;0.9565 &nbsp; &nbsp; 0.1053 &nbsp; &nbsp; &nbsp; &nbsp; Fail
>
>   MiniCPM-o-2.6-8B &nbsp; &nbsp; &nbsp; &nbsp;0.1273 &nbsp; &nbsp; &nbsp; &nbsp; 0.1304 &nbsp; &nbsp; &nbsp; &nbsp; 0.8182 &nbsp; &nbsp; &nbsp;0.7391 &nbsp; &nbsp; 0.0526 &nbsp; &nbsp; &nbsp; &nbsp; Fail
>
> We sincerely appreciate your invaluable feedback and look forward to receiving any further comments or inquiries. We are dedicated to refining this work based on your respected suggestions and appreciate your guidance.
>
> **References**
>
> [1] Zijian Chen, Tingzhu Chen, Wenjun Zhang, and Guangtao Zhai. Obi-bench: Can lmms aid in study of ancient script on oracle bones? arXiv preprint arXiv:2412.01175, 2024.
>
> [2] Sara Ghaboura, Ketan More, Ritesh Thawkar, Wafa Alghallabi, Omkar Thawakar, Fahad Shahbaz Khan, Hisham Cholakkal, Salman Khan, and Rao Muhammad Anwer. Time travel: A comprehensive benchmark to evaluate lmms on historical and cultural artifacts, 2025

---

> > ### Comment · Reviewer_3215 · 2025-08-05
> >
> > Thank the authors for their rebuttal. My concerns have been largely addressed. I am now leaning positive and will raise my rating.

---

> > > ### Author Response · Authors · 2025-08-05
> > > **Response to Reviewer 3215**
> > >
> > > We sincerely thank you for your thoughtful comments and constructive questions. We are especially grateful for your engagement during the rebuttal stage and truly appreciate your willingness to reconsider and raise the rating.

---

### Note · Authors · 2025-08-16

We appreciate the ACs, SACs and all reviewers for your time and constructive feedback, which have significantly strengthened our work.

**Summary of Key Strengths**

We are encouraged by the recognition of our work’s contributions and highlight the following key strengths acknowledged across the reviews:

- **Novelty, quality, and task diversity**: MS-Bench is grounded in scholarly practice, with archaeological workflows and real-world hierarchical task design.
- **Thorough and systematic evaluations**: The rigorous evaluation of 32 LMMs across four prompting strategies yields critical and comprehensive insights.
- **Clear presentation and usability**: The writing, figures, tables, and comprehensive supplementary materials were commended for their clarity and accessibility.


**Key Clarifications and Supplementary Work**

To address the concerns raised by the reviewers, we conducted substantive clarifications and additional experiments, summarized as follows:

- **Applicability to Other Cultures and Manuscripts**: We focused on the Dunhuang corpus—our most accessible and well-curated source, supported by collaborations with domain experts—as a first concrete step to construct high-quality tasks grounded in archaeological workflows and enable rigorous evaluation. We position MS-Bench as a reusable and extensible framework that offers transferable task structures and evaluation protocols applicable to other manuscript traditions in future work.

- For Reviewer **3215**, we expanded culture-aware evaluation metrics to better assess high-level cultural understanding tasks.

- For Reviewer **dWUb**, we further investigated the performance anomaly in allograph normalization and conducted additional ablation studies on prompting strategies across task types. We proposed prompting recommendations and provided actionable insights for archaeologists and future benchmark designers. We also enhanced the dataset and codebase to facilitate practical reuse by both AI and domain experts.

- For Reviewer **Ynz3**, we enriched dataset statistics and included illustrative, case-based analyses to clarify the varying impacts of CoT and V-RAG across tasks.

We will incorporate these clarifications, analyses, and improvements to make the paper more informative and accessible for both AI and archaeology researchers. We sincerely appreciate the thoughtful feedback and engagement from all reviewers.

---

### Decision · Program_Chairs · 2025-09-18

**Decision:**

Accept (poster)

**Comment:**

This paper presents a dataset for studying ancient manuscripts (specifically from Dunhuang).  Multiple foundation models are tested on manuscript understanding tasks such as character recognition and assessment of manuscript damage.

The main strength of this paper is the careful curation of a high quality dataset and rigorous evaluation of existing models.  I believe this paper will both spur interesting modelling research, but also encourage curation of new datasets and research questions.

One major weakness of this work is that it only considers manuscript from Dunhuang.  However, I believe the depth of work done outweighs this.  Some reviewers also mentioned that there could be more exploration into why particular models work (reviewer dWUb "The anomaly noted in section 3.1 of a smaller model overperforming ..." and Ynz3 "The authors should further investigate the underlying reasons for the impact of CoT and V-RAG...").  These concerns to be well address in the author rebuttal.

There was not discussion between reviewers, but all reviewers read the rebuttal and many acknowledged that their questions had been answered.

Overall, I think the weaknesses are outweighed by the strengths.  I also believe other researchers could learn from the careful data curation process so am happy to recommend acceptance for this paper.

===== FINAL UPDATE FROM DB Track PCs ====

The final decision for this paper has been taken by the program chairs after consultation with the SACs. All Senior Area Chairs have ranked papers according to the feedback from the AC during the review process. We decided to leave the original meta-review to reflect the opinion of the AC in light of the initial discussions with reviewers and SAC.